# The HIV-1 envelope protein gp120 is captured and displayed for B cell recognition by SIGN-R1[+] lymph node macrophages

**Chung Park[1], James Arthos[2], Claudia Cicala[2], John H Kehrl[1]\***

[1]B-cell Molecular Immunology Section, Laboratory of Immunoregulation, National Institutes of Allergy and Infectious Diseases, Bethesda, United States; [2]Immunopathogenesis Section, Laboratory of Immunoregulation, National Institutes of Allergy and Infectious Diseases, Bethesda, United States

**Abstract** The HIV-1 envelope protein gp120 is both the target of neutralizing antibodies and a major focus of vaccine efforts; however how it is delivered to B cells to elicit an antibody response is unknown. Here, we show that following local gp120 injection lymph node (LN) SIGN-R1[+] sinus macrophages located in interfollicular pockets and underlying SIGN-R1[+] macrophages form a cellular network that rapidly captures gp120 from the afferent lymph. In contrast, two other antigens, phycoerythrin and hen egg lysozyme, were not captured by these cells. Intravital imaging of mouse LNs revealed persistent, but transient interactions between gp120 bearing interfollicular network cells and both trafficking and LN follicle resident gp120 specific B cells. The gp120 specific, but not the control B cells repetitively extracted gp120 from the network cells. Our findings reveal a specialized LN antigen delivery system poised to deliver gp120 and likely other pathogen derived glycoproteins to B cells.

**\*For correspondence:** jkehrl@niaid.nih.gov

**Competing interests:** The authors declare that no competing interests exist.

## Introduction

The human immunodeficiency virus (HIV-1) functional envelope spike is a trimer of non-covalently associated gp120/gp41 heterodimers, which are coated with N-linked carbohydrates that shield vulnerable protein surfaces from antibody recognition (*Bonomelli et al., 2011*; *White et al., 2011*). The host cell glycosylation pathways attach these carbohydrates (*Varki et al., 2009*). However, the glycosylation processing of gp120 diverges from typical host glycoproteins resulting in densely packed patches of oligomannose glycans (*Doores et al., 2010*; *Bonomelli et al., 2011*). Such clusters do not occur on mammalian glycoproteins and, two such sites on the envelope, one associated with the first/second hypervariable loops (V1/V2-glycan), and the other around the third hypervariable loop (V3-glycan) have served as targets for broadly neutralizing antibodies (*Bonomelli et al., 2011*; *Raska et al., 2014*). The glycan shield protects additional sites of viral vulnerability including the gp120 CD4 binding site and the envelope membrane proximal region (*Raska et al., 2014*). The impact of the glycan shield on the uptake of gp120 by antigen presenting cells (APCs) and its subsequent delivery to B cells in lymph nodes (LNs) or the spleen is unknown.

For B cells to mount an antibody response to an antigen such as gp120 they must encounter intact antigen. Since most B cells reside inside lymphoid follicles in the spleen, LNs, and at mucosal immune sites, most studies of LN antigen delivery have focused on the transport of antigen to the LN follicle and its subsequent loading onto follicular dendritic cells (FDCs) (*Pape et al., 2007*; *Phan et al., 2007*; *Batista and Harwood, 2009*; *Roozendaal et al., 2009*; *Suzuki et al., 2009*; *Cyster, 2010*;

**eLife digest** The human immune system contains many different cell types, which play specific roles in defending the body from invading pathogens, such as bacteria and viruses. For example, macrophages engulf and digest foreign material, whereas specialized B cells termed plasma cells produce molecules called antibodies that help to destroy specific pathogens. However, specific antibodies are only produced if naive B cells have already encountered the pathogen or its surface proteins.

Attempts to improve how the immune system responds to the human immunodeficiency virus (HIV-1) have failed to control and prevent infection. One of the main components of many prospective HIV-1 vaccines is a protein called gp120, which is located on the surface of the virus. Specific B cells recognize this protein and can develop into plasma cells that produce antibodies against HIV-1. However, little is known about how these specific B cells initially get exposed to gp120.

Park et al. injected gp120 into mice, and used sophisticated microscopy to track its movement through the animal. This revealed that gp120 is rapidly transported to nearby lymph nodes—organs that are spread throughout the body, and play an important role in maintaining the immune response. Specialized macrophages can then capture and deliver gp120 to other macrophages in the lymph node.

These specialized macrophages serve as a gp120 reservoir and are located in part of the lymph node that is a bit like a traffic hub, in that other immune cells constantly pass through it. As such, B cells that specifically recognize gp120 have a high likelihood of encountering these gp120-bearing macrophages, thereby allowing the specific B cells to extract gp120, develop into plasma cells, and produce HIV-1 specific antibodies. Manipulating this macrophage network may help to optimize the antibody responses to gp120 and so, in the future, could provide a way of treating or preventing HIV-1 infections.

*Yuseff et al., 2013*). FDCs retain antigen and are needed for the clonal selection of B cells with high affinity antigen receptors during germinal center reactions. Following local injection most antigens access the afferent lymph and are rapidly transported into the subcapsular sinus of the regional LN. Hen egg lysozyme (HEL) is a low molecular weight protein that can rapidly access LN follicle via the conduits (*Roozendaal et al., 2009*). The conduits are an interconnected network of tubules that function as a molecular sieve allowing fluid and small molecules to enter the LN from the subcapsular sinus (*Gretz et al., 1997*). However, gp120 is too large to enter the conduits as is phycoerythrin (PE), a fluorescent non-glycosylated algae protein, whose delivery to FDCs has been examined as an antigen–antibody complex (*Phan et al., 2007*). PE immune complexes are efficiently trapped by subcapsular sinus macrophages (SSMs) and delivered to FDCs in a complement dependent manner. Furthermore, cognate B cells residing in the follicle can acquire the antigen directly from the overlying SSMs. Keyhole limpet hemocyanin (KLH) is perhaps a better model for gp120 as it also heavily glycosylated, but similar to PE, KLH has been studied as an immune complex (*Roozendaal et al., 2009*). While these studies have contributed to our understanding of FDC loading and germinal center responses, the kinetics of primary antibody responses do not favor naïve, recirculating B cell encountering high molecular weight antigens on FDCs (*MacLennan, 2007*). This suggests that another mechanism tailored to deliver a neo-antigen such as gp120 to cognate, naïve B cells might exist.

One possibility is the SSMs that directly overlie the LN follicle. SSMs are CD169$^+$CD11b$^+$F4/80$^-$ and besides capturing immune complexes they also retain particulate material such as ferritin and liposomes (*Gray and Cyster, 2012*). Perhaps less likely are two other types of LN macrophage, medullary sinus macrophages (MSMs) and medullary cord macrophages (MCMs). Like the SSMs, the MSMs are also CD169$^+$CD11b$^+$, but they also express F4/80 and the pattern recognition receptors SIGN-R1 and MARCO. Their known functions are to clear particulates, pathogens, and dying cells (*Gray and Cyster, 2012*). The MCMs are CD169$^-$CD11b$^+$F4/80$^+$April$^+$ and they support plasma cell homeostasis. However, these macrophages are not very motile and localized far from most follicular and trafficking B cells. A better candidate is the interfollicular macrophages (IFMs) (*Gray and Cyster, 2012*). They are

phenotypically similar to the MSMs, but they reside between the LN follicles in the interfollicular channel (IFC), a site where early T-B cell collaboration occurs (*Kerfoot et al., 2011*). High endothelial venules (HEVs) and cortical sinus lymphatics are located nearby (*Park et al., 2009*). However, the functional role of IFMs in humoral immunity is poorly defined. LN resident dendritic cells (DCs) predominately sample the conduit contents making them an unlikely contender; however DCs in the vicinity of locally administered antigens can capture them, enter the afferent lymphatics, and access local LNs via the IFCs (*Qi et al., 2006*). Yet such a mechanism is slow compared to the rapid antigen delivery via the lymph. Local DC-mediated antigen delivery is likely important for those antigens that do not enter the afferent lymph.

To test how gp120 is captured in the LN we injected mice with fluorescently labeled gp120 near the inguinal LN. We followed the label using thick LN sections and confocal microscopy, and by intravital two-photon laser scanning microscopy (TP-LSM). We also developed a gp120 overlay assay that allowed the identification of gp120 binding cells in lymphoid organ sections. To determine how cognate B cells acquire gp120 we adoptively transferred B cells from mice in which the variable portions of the human b12 neutralizing antibody were introduced into endogenous mouse Ig heavy and light chain loci by gene targeting (*Ota et al., 2013*). The b12 antibody recognizes the CD4 binding site in gp120 (*Burton et al., 1991*; *Roben et al., 1994*). Following injection of fluorescently labeled gp120 we could track the acquisition of antigen by the gp120 specific B cells using intravital TP-LSM. Together these studies identified a group of macrophages that overlie the IFC and which extend to the cortical ridge and sinuses that bound and delivered gp120 to both re-circulating and follicle B cells. These IFMs are adjacent to, but distinct from the SSMs that overlie the LN follicle. We also identified a SIGN-R1 positive cell located in the splenic marginal zone that rapidly acquired blood borne gp120. Our studies revealed an efficient mechanism for exposing trafficking naïve B cells to gp120.

## Results

### Locally injected gp120 is captured in the LN by SIGN-R1$^+$ subcapsular macrophages and SIGN-R1$^+$ IFC macrophages

For these studies we used an early HIV-1 viral isolate subtype A/C gp120, R66M, expressed in 293F cells (*Nawaz et al., 2011*), and injected 1 μg of fluorescently labeled gp120 near the base of the mouse tail. Confocal microscopy of thick LN sections prepared 2 hr after gp120 injection revealed that labeled gp120 had been captured by LN macrophages that overlie and extend into IFCs and that localize at the cortical medullary junction (*Figure 1A*, top and middle panels). The asymmetric gp120 signal results from the gp120 accessing the afferent lymphatics serving the left side of the inguinal LN as it is oriented in the figure. Further immunostaining revealed that gp120 co-localized with SIGN-R1 (*Figure 1A*, bottom panel), a c-type lectin and functional ortholog of DC-specific ICAM-3-grabbing non-integrin (DC-SIGN), which has been implicated in HIV-1 transmission by human DCs (*Geijtenbeek et al., 2000*; *Kang et al., 2003*). Intravital TP-LSM revealed the rapid appearance of gp120 in the subcapsular sinus and identified the same subset of SIGN-R1$^+$ macrophages capturing gp120 (*Figure 1B*). Together the imaging and flow cytometry identified the gp120 binding cells as SIGN-R1$^+$ CD169$^{mid}$CD11b$^{mid}$CD4$^+$/CD11c$^-$F4/80$^{low}$ sinus macrophages (SIGN-R1$^+$ subcapsular macrophages) and SIGN-R1$^+$/CD169$^{mid}$CD11b$^{low}$CD4$^+$CD11c$^-$F4/80$^{low}$ IFM (SIGN-R1$^+$ IFC macrophages) (*Figure 1C*). These cells are to be distinguished from the SIGN-R1$^+$CD11b$^+$ DCs located in the medullary region, previously identified to uptake inactivated influenza virus (*Gonzalez et al., 2010*). The SIGN-R1$^+$ DCs also bound gp120 and are likely important for T cell priming (*Figure 1D*). Next, we investigated the role of SIGN-R1 in gp120 binding. To do this we first checked whether gp120 bound in vitro to LN SIGN-R1$^+$CD169$^{mid}$CD11b$^+$ cells and whether unlabeled gp120 competitively inhibited the binding. We found that gp120 bound a phenotypically similar subset of macrophages and that unlabeled gp120 reduced the binding of the labeled material (*Figure 1E*). When we added a SIGN-R1 blocking antibody with the labeled gp120, the level of SIGN-R1 on the LN SIGN-R1$^+$CD169$^{mid}$CD11b$^+$ cells declined as did the fluorescent gp120 binding arguing that SIGN-R1 directly participated in the binding (*Figure 1E*). The percentage of cells that bound gp120 declined by approximately 50% in the presence of the SIGN-R1 antibody. To directly visualize these cells in vitro, we sorted Gr-1$^-$CD11c$^-$ SIGN-R1$^+$CD11b$^+$CD169$^+$ cells from mice previously injected with fluorescent gp120. The sorted cells were imaged (*Figure 1—figure supplement 1*). Because of the rarity of the cells in the LN

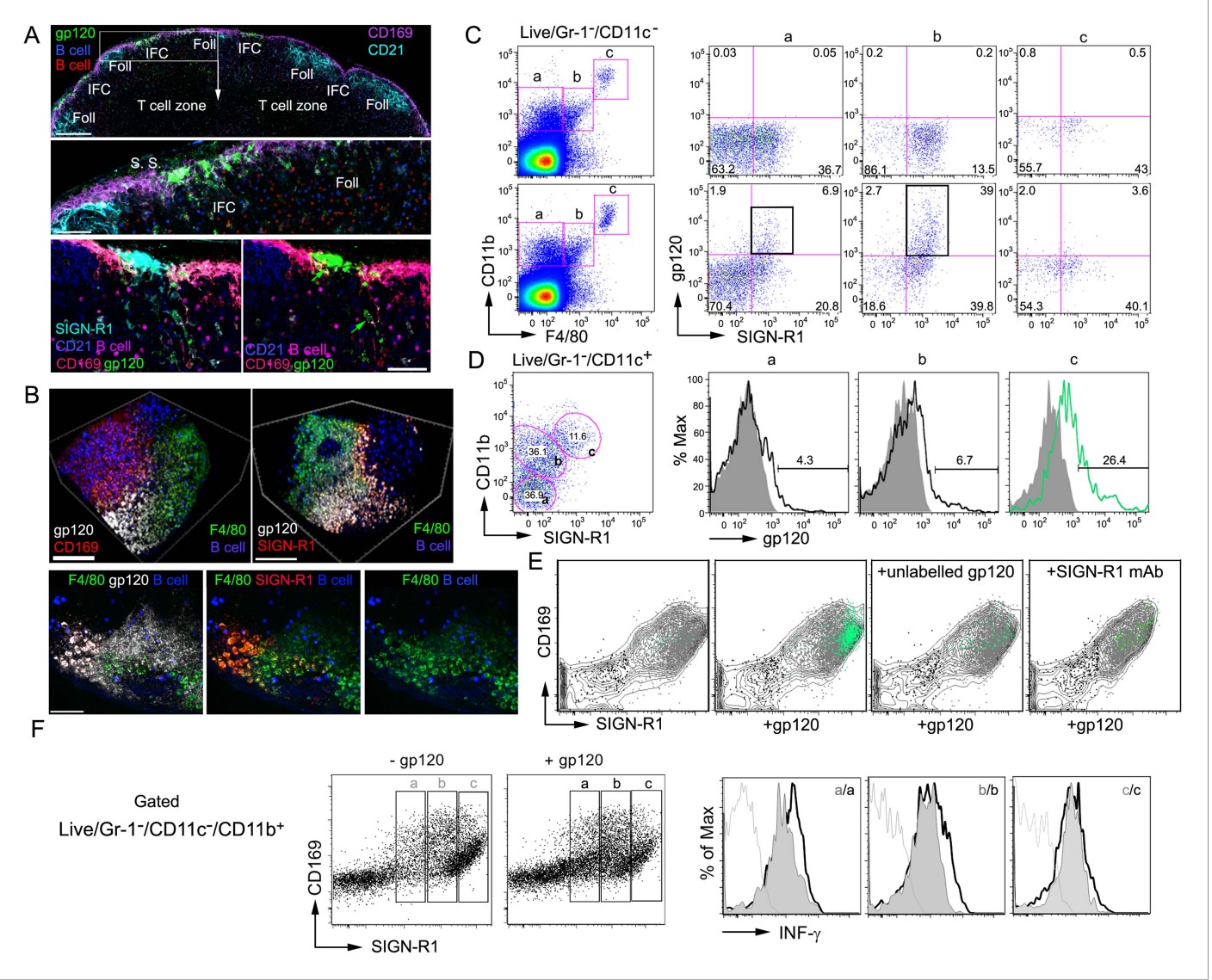

**Figure 1**. SIGN-R1 positive interfollicular channel (IFC) and cortical medullary junction macrophages rapidly accumulate lymph borne gp120. (**A**) Confocal microscopy of thick lymph node (LN) sections prepared from mice that had received adoptively transferred B cells (previous day), injected with fluorescently labeled gp120, and immunostained as indicated. LN section image (sagittal, tiled) shows gp120, green; CD169, pink; CD21/35, cyan; and B cells, red and blue. Scale bar is 200 μm (top). A zoomed image of the white boxed area is shown. Scale bar is 60 μm (middle). Images of an IFC are shown: gp120, green; CD169, red; SIGN-R1, cyan; CD21/35, blue; adoptively transferred B cells, pink; and CD169, red (bottom, left). SIGN-R1 signal removed (bottom, right). Arrows indicate gp120 positive cells. Scale bars is 50 μm. (**B**) Intravital two-photon laser scanning microscopy (TP-LSM) images of the inguinal LN from a mouse injected with fluorescent gp120 and the indicated antibodies. The top images over the IFC show gp120, white; CD169, red; F4/80, green; and adoptively transferred B cells, blue, (left panel). SIGN-R1 antibody, red, used instead of CD169 (right panel). Scale bars are 100 μm. The bottom images are from the follicular-medullary junction and show gp120, white; SIGN-R1, red; F4/80, green; and adoptively transferred B cells, blue. Scale bar is 50 μm. (**C**, **D**) Flow cytometry of LN cells immunostained and gated as indicated using inguinal LN cells from a mouse injected with fluorescent gp120 1.5 hr previously, or not. LiveGr-1⁻CD11c⁻ gated population plotted for F4/80 vs CD11b. Gates 'a', 'b', and 'c' as indicated were re-plotted to show SIGN-R1 vs gp120 in right three plots (top 2 rows). (**C**). LiveGr-1⁻CD11c⁺ population is shown plotted for SIGN-R1 vs CD11b. Histogram of indicated three populations (a, b, and c) plotted as gp120 signal (black line) vs % of maximum intensity compare to gp120 negative control (shaded). Numbers are % gp120 positive cell population in gate (**D**). (**E**) In vitro binding by LN cells incubated with fluorescent gp120, or not, and in the presence of non-labeled gp120 or non-labeled SIGN-R1 antibody (different epitope) and then analyzed by flow cytometry. LiveGr-1⁻CD11c⁻CD11b⁺ cells were analyzed for gp120 vs SIGN-R1 (not shown) and the CD11b⁺ cells, gray contour; the SIGN-R1⁺gp120⁺ cells, green dots; SIGN-R1⁻gp120⁻ cells, black dots; and SIGN-R1⁺gp120⁻ cells, gray dots, were plotted to show CD169 vs SIGN-R1. (**F**) Interferon-γ intracellular flow cytometry of cells prepared from the inguinal LNs of mice administered gp120 near the tail base, or not, 3 hr prior to collection. LiveGr-1⁻CD11c⁻CD11b⁺ cells were analyzed for SIGN-R1 vs CD169 and separated into three populations (left panels). The levels of intracellular interferon-γ are shown as histograms of maximum intensity in cells from the gp120 non-exposed (gray) and gp120 injected mice (white, outlined by black lines). Unstained control is delineated by a gray line.

*Figure 1. continued on next page*

*Figure 1. Continued*

The following figure supplements are available for figure 1:

**Figure supplement 1**. Sorted SIGN-R1+ macrophages capture gp120.
**Figure supplement 2**. The injection of gp120 triggers transcription of interferon- γ in SIGN-R1+ macrophages as assessed by using a interferon-γ-eYFP reporter mouse.
**Figure supplement 3**. The injection of gp120 triggers transcription of interferon- γ in SIGN-R1+ macrophages.

population the sorted cells were contaminated with other cell types yet many gp120+SIGN-R1+ cells could be visualized. We also cultured the sorted cells with M-CSF. At day 7 the cultured cells were incubated with fluorescent gp120 and immunostained for SIGN-R1. The majority of the cultured cells retained SIGN-R1 expression and most of these cells bound gp120 (*Figure 1—figure supplement 1*). To determine whether the uptake of gp120 triggered a biologic response in the IFC macrophages we injected gp120 locally and checked the intracellular interferon-γ levels in these cells (*Figure 1F*). Some of the LN SIGN-R1+CD169midCD11b+ cells had an elevated level of intracellular interferon-γ compared to control cells. We verified these results using an interferon-γ eYFP reporter mouse (*Reinhardt et al., 2009*). Flow cytometry was used to assess the percent of eYFP positive cells in the gated SIGN-R1+ macrophages (*Figure 1—figure supplement 2*), and to examine the induction of eYFP expression in various other cell populations in the immunized LN (*Figure 1—figure supplement 3*). To verify that the eYFP signal arose from the SIGN-R1+ macrophages we sorted the Gr-1−CD11c−SIGN-R1+CD11b+CD169+ cells and imaged them. We could readily identify SIGN-R1+eYFP+ cells, while the other contaminating cells present in the sorted population lacked YFP expression. Finally, we injected non-labeled gp120 near the inguinal lymph of the reporter mouse and 6 hr later made thick LN sections from the draining LN node and from a distant LN. Confocal microscopy revealed eYFP positive SIGN-R1+CD169mid cells in the IFC region of the draining LN, but similar cells were not present in the distant LN (*Figure1—figure supplement 3*). Together these results identified a subset of mouse subcapsular macrophages that overlie and reside in the IFC, which express SIGN-R1 and rapidly uptake gp120. In addition, our data indicates that the local injection of gp120 likely elicits interferon-γ production by these cells.

## SIGN-R1+ IFC macrophages bearing gp120 contact nearby lymphocytes and SIGN-R1+ splenic marginal zone cells capture blood borne gp120

The IFC connects the subcapsular sinus to the cortical ridge at the boundary of the B and T cell zones. By 3 hr after injection gp120 labelled processes extended into the IFC making contact with B and T lymphocytes (*Figure 2A*). In addition, IFC DCs could be found associated with the SIGN-R1+ IFC cells bearing gp120 (*Figure 2—figure supplement 1*). The likely involvement of SIGN-R1 in the uptake of gp120 by the LN macrophages prompted us to examine whether the SIGN-R1 positive cell known to reside in the marginal zone region of the spleen could uptake gp120 from the blood (*Kang et al., 2003*). Whether these SIGN-R1 positive marginal zone cells are macrophages or resident DCs has been debated (*Lyszkiewicz et al., 2011*), however they are generally referred to as SIGN-R1+ marginal zone macrophages. Following gp120 injection into the blood we observed a subset of marginal zone cells that rapidly acquired the gp120 signal (*Figure 2B*). They expressed high levels of SIGN-R1 and lacked CD11c (data not shown). Confocal microscopy of thick spleen sections immunostained with CD169, CD21, and SIGN-R1 and overlaid with fluorescent gp120 identified a similar marginal zone SIGN-R1 positive cell (*Figure 2C*). A tiled confocal image of a portion of the spleen from the gp120 overlay is also shown (*Figure 2D*), which demonstrates a remarkable overlap between the SIGN-R1 and gp120 signals. These results indicate that there is a network of SIGN-R1 positive macrophages in the LN IFC that provide a platform for nearby B cells and DCs to acquire gp120 and that a subset of SIGN-R1+ cells in the marginal zone of the spleen are also poised to deliver gp120 to splenic marginal zone and trafficking follicular B cells.

The ability to specifically detect gp120 binding cells using an overlay assay prompted us to determine whether we could detect similar macrophages in human LN. To identify the LN follicles and T cell zone we immunostained a LN section for CD19 and CD4 expression. Using 2 adjacent sections

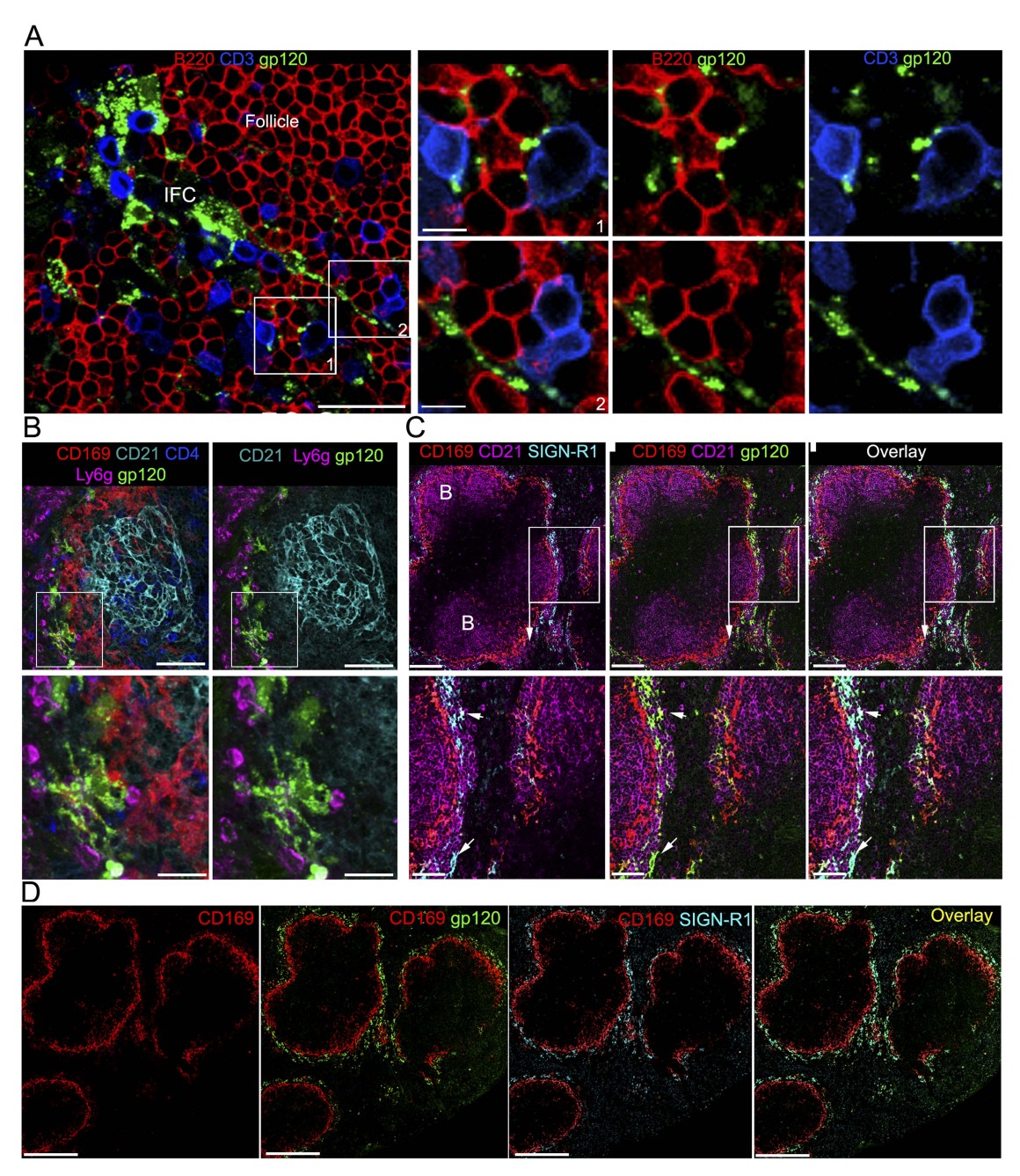

**Figure 2**. IFC cell processes bearing gp120 directly contact B cells and a subset of splenic marginal zone cells also bind gp120. (**A**) Confocal microscopy of a thick LN section from a mouse previously injected with fluorescent gp120 and immunostained for B220 and CD3. Scale bar is 30 μm. Boxed areas in left image were enlarged and shown in the right panels. Scale bars are 10 μm. (**B**) Confocal microscopy of a thick splenic section from a mouse previously injected intravenously with fluorescent gp120 and immunostained with the indicated markers. Zoomed images shown below. Scale bars are 100 μm, above, and 30 μm, below. (**C**) Confocal microscopy of a thick splenic section overlaid with fluorescent gp120 and immunostained for the indicated markers. Zoomed images are shown below. Scale bars are 100 μm, above, and 40 μm, below. (**D**) Tiled confocal microscopy images of a spleen section immunostained with CD169 (red) and SIGN-R1 (cyan) and overlaid with fluorescent gp120 (green). As indicated the images show CD169 alone, CD169 and SIGN-R1; CD169 and gp120; and overlay of all three. Scale bar is 300 μm.

The following figure supplements are available for figure 2:

**Figure supplement 1**. The IFC network macrophages contact CD11c positive cells.

**Figure supplement 2**. Overlay of gp120 visualizes DC-SIGN[+]/CD163[+] macrophages in a human LN section.

we immunostained one for CD163, a human macrophage marker (*Martens et al., 2006*), CD11c, and DC-SIGN; and the other for gp120 and DC-SIGN. This allowed the identification of a group of CD163⁺, DC-SIGN⁺, and CD11c⁻ cells near the LN follicle that bound the overlaid gp120 (*Figure 2—figure supplement 2*). These results indicate that in human LN DC-SIGN expressing macrophage near the follicle may uptake gp120 similar to the mouse IFC SIGN-R1⁺ macrophages.

## The SIGN-R1⁺ subcapsular macrophages uptake gp120 prior to the SIGN-R1⁺ IFC macrophages

To determine the kinetics of gp120 uptake by SIGN-R1⁺ subcapsular macrophages and the underlying SIGN-R1⁺ IFC macrophages, we intravitally imaged for 3.5 hr following gp120 injection. The amount of gp120 associated with the SIGN-R1⁺ subcapsular macrophages gradually increased and then declined, while the underlying network cells incrementally increased their gp120 binding eventually surpassing the sinus macrophages (*Figure 3A,B*, *Videos 1, 2*). Cellular processes labeled with gp120 were visualized extending into the LN follicle. In one instance an endogenous cell (weakly fluorescent) approached and contacted a gp120 labeled cellular process (*Video 3*). The gp120 identified IFC cellular network extended to the cortical ridge and even into the cortical sinuses (*Figure 3C*). These results reveal the rapid gp120 loading of the SIGN-R1⁺ IFC macrophages and suggest that a transport mechanism may exist to transfer gp120 from the superficial to the underlying cells in the IFC.

## The SIGN-R1⁺ macrophage network does not uptake HEL or PE

We checked the specificity of this cellular network by comparing gp120 to two other proteins; Alexa-488 labeled HEL, which because of its low lower molecular mass should enter the conduits (*Roozendaal et al., 2009*), and 4-Hydroxy-3-nitrophenylacetyl (NP) modified PE, a large non-glycosylated fluorescent algae protein. NP-PE predominately targeted the sinus lining cells while as expected Alexa-488-HEL largely entered the conduit system (*Figure 4A,B*). The first region of interest (ROI-1) overlies the LN follicle and shows the sinus lining cells have taken up NP-PE, while the same cells have little gp120 (*Figure 4B*). Conduits filled with Alexa-488-HEL (blue), which lack gp120 and NP-PE, penetrated into the LN follicle. The ROI overlying the IFC (ROI-2) shows strong gp120 positivity, some sinus lining cells are NP-PE positive, but appear distinct from the gp120 positive cells (*Figure 4B*). This is evident from the analysis of the two ROI defined within ROI-2. The third, ROI-3, connects the IFC to a lymphatic sinus. Numerous solely gp120 positive cells are present, while conduits and likely fibroreticular and lymphatic cells are outlined by HEL uptake. Again we detected little overlap between NP-PE bearing cells and those bearing gp120 (*Figure 4B*). Long cellular processes labeled by the presence of fluorescent gp120 can be visualized extending into the IFC making contact with other cells (*Figure 4C*). Three-dimensional reconstruction of the imaging data shows that gp120 resides both within and on the surface of the IFC macrophages (*Figure 4C*, right panel). The IFC macrophage processes and gp120 signal often wrapped around the conduits and fibroreticular cells (*Figure 4D,E*). These data highlight three different mechanisms of antigen uptake by LN cells, which depend upon the size and composition of the antigen.

## The carbohydrate structure of gp120 influences the uptake of gp120 by the SIGN-R1⁺ macrophage network, while trimeric gp120 is captured similar to monomeric gp120

SIGN-R1 selectively recognizes α-2,6-sialylated glycoproteins (*Silva-Martin et al., 2015*). Oligosaccharides present on the envelope of various viruses including HIV contain terminal α-2,6 sialic acid linkages. To assess whether differences in the carbohydrate structure of gp120 affected its uptake by this cellular network we labeled CHO-S or 293F expressed R66M gp120 with Alexa-488 (green) or Alexa-594 (red). CHO-S expressed proteins have a heterogeneous pattern of oligo-mannose and complex carbohydrate type glycans while 293F expressed proteins have mostly complex carbohydrate type glycans. Furthermore, 293F cells sialylate the terminal galactose moieties of complex carbohydrates using −2,3 and −2,6 linkages while CHO-S cells only use the −2,3 linkage (*Nawaz et al., 2011*). By injecting both labeled gp120 simultaneously we could follow and compare the acquisition of gp120 by the network cells. We found both gp120 preparations accessed the previously defined cellular network; however, the 293F derived gp120 bound the SIGN-R1⁺ IFC macrophages better than did the CHO-S

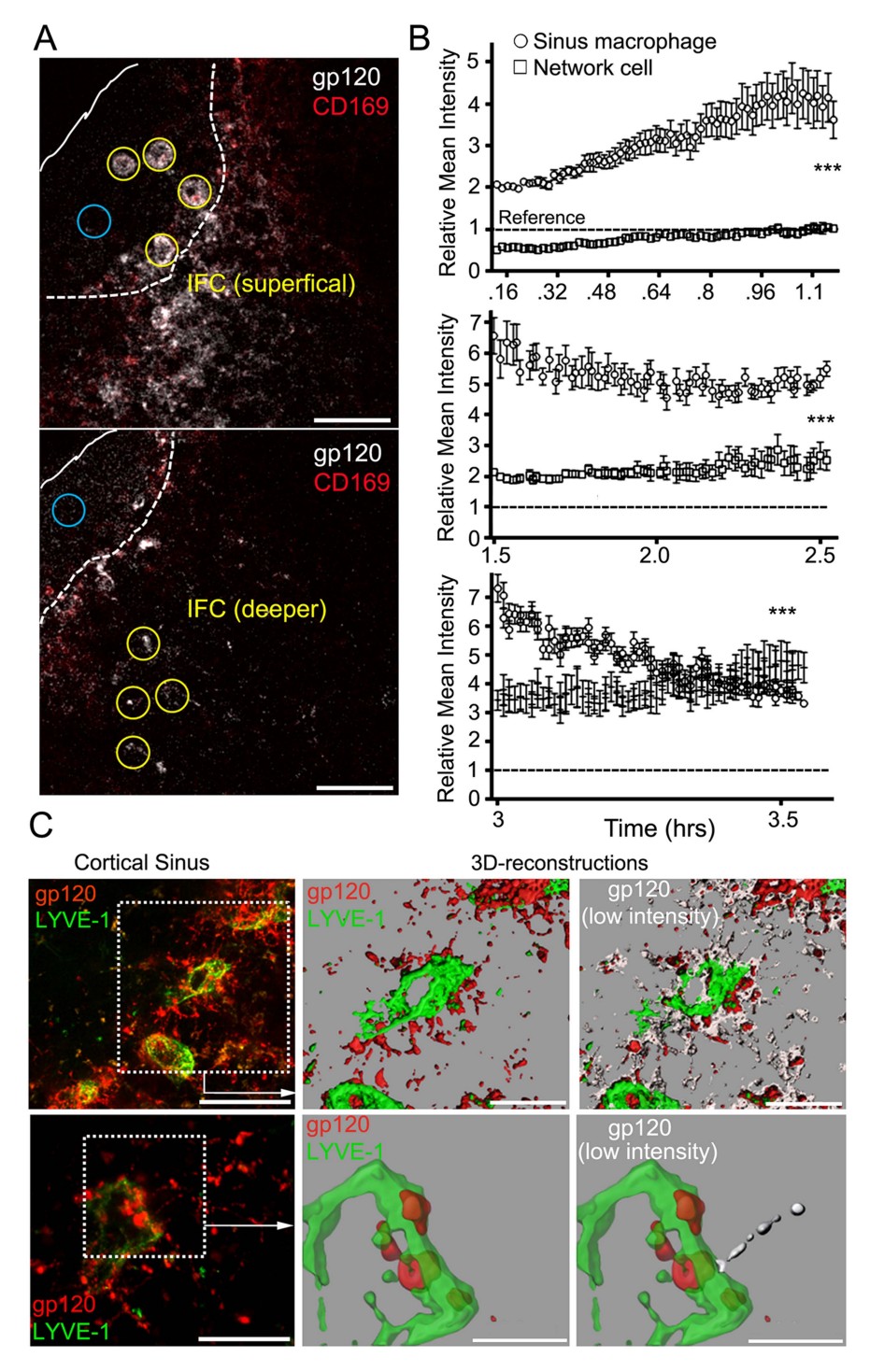

**Figure 3**. The IFC network of macrophages extends from the subcapsular sinus to the cortical sinus. (**A**) Intravital TP-LSM images of the inguinal LN 8 min after injection of fluorescent gp120 and 30 min after CD169. Yellow circles indicate sinus macrophages (upper panel) or network cells in the IFC (lower panel). Blue circles indicate subcapsular sinus lumen. Distance between two slices is 30 μm. Scale bars are 50 μm. (**B**) The gp120 signal associated with SIGN-R1+ SM (O) or deeper in the IFC (□) was quantitated over time following gp120 injection: 8 min-1 hr 14 min, upper; 1 hr 30 min–2 hr 30 min, middle; and 3 hr–3 hr 32 min, bottom panel. Calculated slopes (O,□) are 1.59 ± 0.046 and 0.45 ± 0.013, upper; −1.04 ± 0.16 and 0.63 ± 0.11; middle, and −5.61 ± 0.23 and 2.35 ± 0.33; bottom panel. In the same panel the slopes differed by a p value <0.0001. The reference signal of gp120 in subcapsular

*Figure 3. continued on next page*

*Figure 3. Continued*

sinus lumen is shown with dotted lines. Error bars, ±SEM (**B**). (**C**) Intravital TP-LSM images from deep in the IFC following injection of fluorescent gp120 and LYVE-1 antibody. The left upper panel show an image from a 30 μm z-projection. Scale bar is 100 μm. Dotted box was reconstructed using the LYVE-1 signal, green, and gp120 signal, red (upper middle panel) and modified by adding low intensity gp120 signal, white (upper right panel). Scale bars are 50 μm. The left lower panel shows a higher power 20 μm z-projection image. Scale bar is 50 μm. Dotted box was reconstructed using the LYVE-1 signal, green, and gp120 signal, red (lower middle panel) and modified by adding the low intensity gp120 signal, white (lower right panel). Scales bars are 20 μm.

derived gp120, irrespective of how it was labeled (*Figure 5A–C*). Perhaps as a consequence the antibody response to 293F derived gp120 exceeded the response to CHO-S gp120 (*Figure 5D*).

To check and compare the FDC network loading of the two gp120 preparations we waited 3 weeks after boosting mice previously immunized with 293F gp120 and injected both 293F and CHO-S expressed gp120 near the inguinal LN. Using intravital TP-LSM we checked both the loading of the IFC macrophages and the FDC network in a nearby follicle. As we had previously noted the IFC macrophages rapidly captured the lymph borne gp120, however, in contrast to the naïve mouse within 4 hr of injection both gp120 preparations had loaded on to the FDC network of a germinal center (*Figure 5E*). The germinal center region was outlined by adoptively transferred naïve B cells. The strongly fluorescent cells in the germinal center are likely tingible body macrophages.

To more drastically change the carbohydrate structure of gp120, we treated the purified 293F gp120 with PNGaseF, an amidase that cleaves between the innermost GlcNAc and asparagine residues of high mannose, hybrid, and complex oligosaccharides. The untreated gp120 and the PNGaseF treated gp120 were used in a mouse LN overlay assay. As previously the untreated gp120 bound strongly to the SIGN-R1 positive macrophages, while the PNGase F treated gp120 exhibited much less selectivity, binding to many different cell types. This resulted in a reduction in the co-localization between SIGN-R1 and gp120. We verified that the PNGase F treatment appropriately altered the molecular mass of gp120 (*Figure 5—figure supplement 1*). Since the trimeric version of gp120 offers different antigenic determinants and is rapidly becoming accepted as the preferred

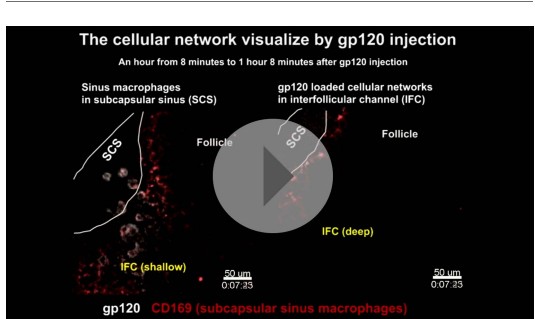

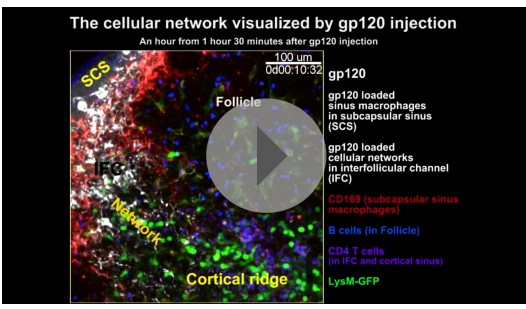

**Video 1.** Intravital two-photon laser scanning microscopy (TP-LSM) images of the interfollicular channel (IFC) network that extends from the subcapsular sinus to the cortical sinus. Images from two focal planes separated by 30 μm and located over the IFC channel of a mouse inguinal lymph node (LN). The mouse had previously been injected with fluorescent CD169 antibody, red, which delineates the subcapsular sinus macrophages (SSMs). The images were acquired over an hour (8 min–1 hr 8 min post fluorescent gp120, white, injection into the tail base). The white lines delineate the subcapsular sinus. Scale bar is 50 μm. Time counter shows hr:min:s.

**Video 2.** Intravital TP-LSM images of the gp120 loaded cellular network in the IFC of the inguinal LN. An image sequence of a 30 μm z-projection was acquired from a LysM-EGFP mouse, which had previously received by adoptive transfer both B cells, blue, and CD4 T cells, purple. Host endogenous neutrophils/monocytes, strong green signal, and stromal cells, weak green signal, can be seen on the basis of their expression of LysM-EGFP. Images were acquired for an hr from 1.5–2.5 hr after fluorescent gp120, white, injection near the tail base. GFP positive cells can be seen flowing in blood vessels in the IFC. CD169 antibody, red, delineated the SSMs. Second harmonic signal, blue, from collagen delineated LN capsule. Scale bar is 100 μm. Time counter shows hr:min:s.

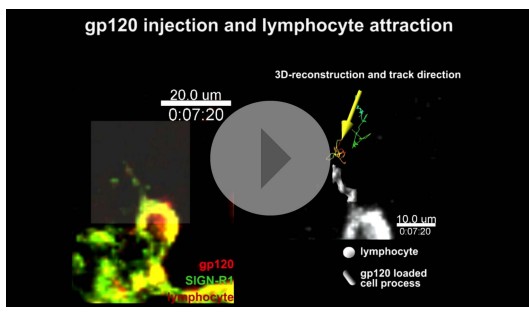

**Video 3.** Intravital TP-LSM images of the dynamic interaction of SIGN-R1[+] gp120[+] macrophages and a lymphocyte. An image sequence of a 20 µm z-projection was acquired from the inguinal LN following nearby injection of fluorescent gp120, red, and SIGN-R1, green, antibody (left). Track and displacement (yellow arrow) of lymphocytes (gray spot) superimposed with 3D-reconstructed images of cell process is gray (right). Scale bars are 20 µm and 10 µm. Time counter shows hr:min:s.

vaccine candidate (*Sanders et al., 2013*), we also checked whether a trimeric version of gp120 was captured by the SIGN-R1[+] macrophages overlying the IFC. We found a very similar pattern of uptake as we had observed with the monomeric gp120 (*Figure 5—figure supplement 2*). Together these results indicate that the carbohydrate structure of gp120 can affect the binding of gp120 to the SIGN-R1 positive macrophages, which may affect the subsequent antibody response. However minor differences in the glycosylation of gp120 did not impact FDC loading in the setting of gp120 specific antibody.

## Newly arriving gp120 specific B cells can capture gp120 from the IFC cellular network

Next, we monitored the interaction of B cells with the gp120 bearing cells in the LN by adoptively transferring b12 knock-in B cells that possess gp120 reactive antigen receptors. The knock-in B cells bearing the H & L, H chain, and L chain genes bind gp120 with high, low, and no detectable affinity, respectively (*Burton et al., 1991*; *Ota et al., 2013*). Previously 90% of all mature b12 HL B cells bound soluble trimers of HIV Env (the JRFL isolate), as measured by flow cytometry. Similar to the results with the soluble trimers of the JRFL envelope, labeled R66M gp120 bound more than 90% of the splenic follicular and marginal zone b12 HL B cells (*Ota et al., 2013*) (*Figure 6A*). Of note, the average gp120 mean fluorescent intensity on marginal zone B cells exceeded that on follicular B cells by twofold. Next we transferred gp120 specific B cells 2 hr after the injection of gp120 to assess the delivery of gp120 to newly arriving HIV-1 specific B cells. The tracks of the newly arriving b12 HL B cells and b12 L B cells predominately localized in the IFC although the b12 HL B cells focused preferentially on the gp120 bearing network cells (*Figure 6B*). The motility patterns of the b12 HL and b12 L B cells differed. Although the average velocities were similar, the b12 HL B cells moved less straight with more speed variability and exhibited greater displacements than the b12 L B cells (*Figure 6C*). The b12 HL B cells interacted vigorously with the gp120 bearing cells extracting gp120 from the network cells, which accumulated in their uropods (*Figure 6D*, *Video 4*). By loading the b12 HL B cells with the Ca[2+] sensitive dye, Calcium Orange, we could observe transient increases in intracellular Ca[2+] as the b12 HL B cells interacted with, and extracted gp120 from the IFC cells (*Figure 6E*, *Video 5*). The intracellular Ca[2+] rise often occurred in conjunction with an increase in gp120 in the b12 HL B cells (*Figure 6E*). These results indicate that newly arriving recirculating B cells can acquire antigen from the IFC macrophages bearing gp120. Similarly naïve B cells will have the opportunity to encounter cognate antigen on IFC cells as they exit the LN follicle.

## LN follicle gp120 specific B cells can extract gp120 from IFC network cells at the follicle edge

To determine whether B cells resident in LN follicles might also acquire cognate antigen from the IFC network, we adoptively transferred b12 HL and wild type (WT) B cells to recipient mice the day prior to gp120 injection. This allowed the B cells to localize in the follicle. Then, we injected gp120 and over the next 2 hr monitored its uptake by the IFC network and the behavior of B cells in the LN follicle. As noted previously the IFC network cells rapidly accumulated gp120 and within 30 min fine cellular processes bearing gp120 became visible along the follicle edge (*Figure 7A*). Tracking the transferred B cells near the follicle edge revealed that the b12 HL B cells moved slower and tended to remain in the imaging space longer resulting in a longer track lengths and increased displacements (*Figure 7B*). As a consequence b12 HL B cells accumulated at these sites while the control B cells did not (*Figure 7C*). The b12 HL B cells made numerous, transient interactions with the gp120 bearing cellular

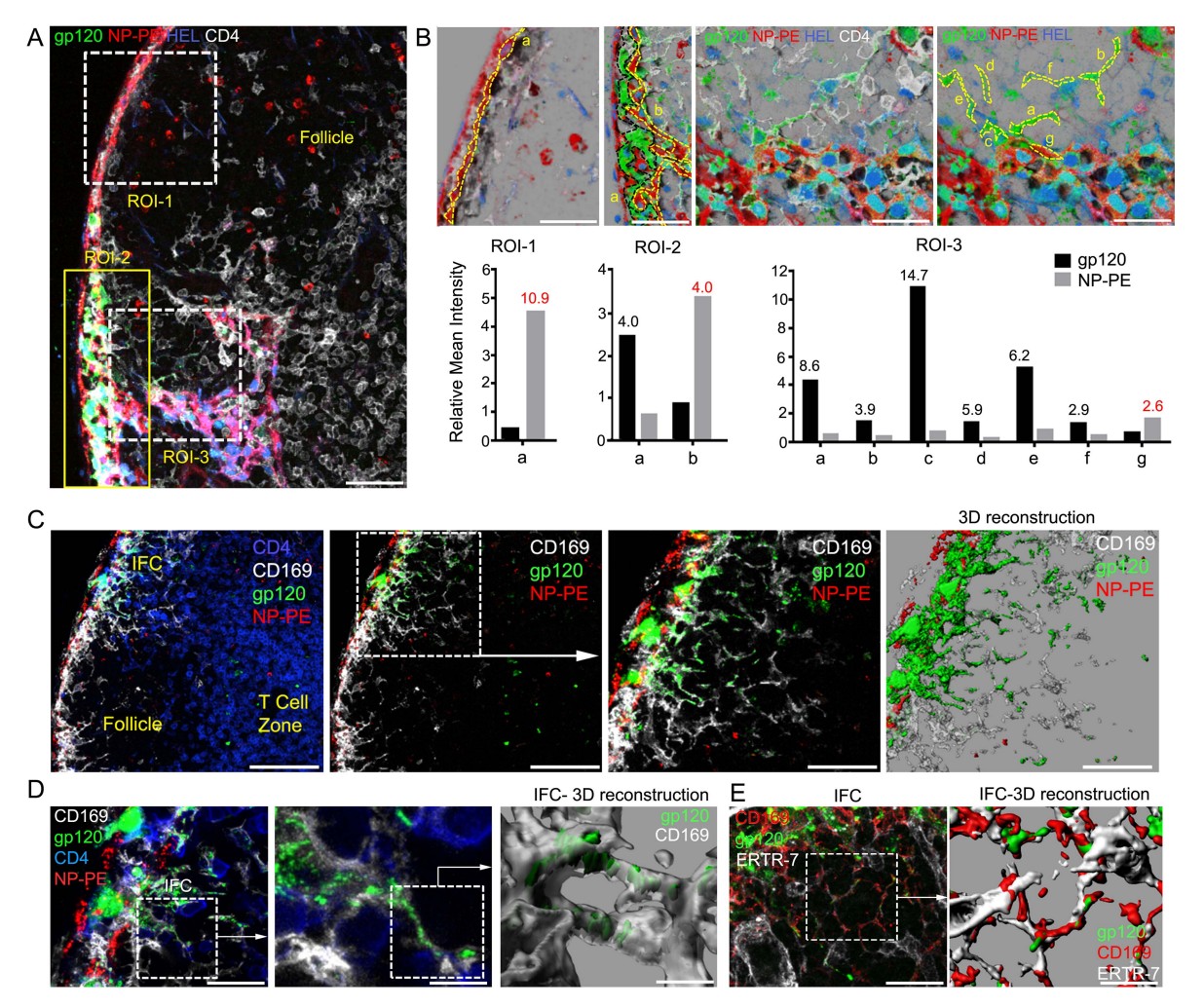

**Figure 4**. IFC network macrophages do not uptake hen egg lysozyme (HEL) or nitrophenylacetyl (NP)-phycoerythrin (PE). (**A**, **B**) Confocal microscopy of a thick LN section immunostained for CD4, white, following injection of NP-PE, red, fluorescent gp120, green, and fluorescent HEL, blue, near the inguinal LN. Three regions of interest are shown over LN follicle region of interest (ROI-1), superficial IFC (ROI-2), and deep IFC (ROI-3). Scale bar is 40 µm (**A**). In part **B** each ROI is further subdivided as indicated by letters to delineate specific cells or groups of cells. The fluorescent intensity of NP-PE or gp120 in each of these regions was quantitated and is indicated. Numbers in graphs indicate fold difference. (**C**) Confocal microscopy image of a thick LN section from a mouse previously injected with fluorescent gp120, green, plus NP-PE, red, and immunostained for CD169, white, and CD4, blue. CD4 is excluded in the 2nd–4th panels. Electronically zoomed image of boxed area is shown in 3rd panel. A 3-D reconstruction of the 3rd panel image is shown in the 4th panel. Scale bars from left to right are 100, 100, 50, and 50 µm. (**D**) Confocal microscopy image of a thick LN section immunostained for CD4 and CD169 prepared from a mouse injected near the inguinal LN with fluorescent gp120 and NP-PE. The middle image is an electronically zoomed image of the region in the left panel. A portion of the middle image was used to perform a 3-D reconstruction of the imaging data shown in the right panel. The scale bars from left to right are 25, 10, and 5 µm. (**E**) Confocal microscopy image of a LN section immunostained for ERTR-7, white, and CD169, red, from a mouse previously injected with fluorescent gp120, green, focusing on the IFC. The indicated portion of left panel was used for the 3-D reconstruction shown in the right panel. The scale bars from left to right are 30 and 15 µm.

processes (*Video 6*). Tracking individual b12 HL B cells revealed that the B cells slowed as they extracted antigen from the network, after which they sped up and the cell associated gp120 signal declined, whereupon they re-engaged the network and extracted more gp120 (*Figure 7D*). Interestingly, the average velocities of both the WT and the gp120 specific B cells increased over time following the gp120 injection (*Figure 7E*). Detailed analyses of six tracks from B cells located in the LN follicle and near the intrafollicular channel are shown (*Figure 8*). These results show that antigen loaded IFC can provide a source of antigen for cognate B cells traversing the edge of the LN follicle and perhaps provide signals that enhance B cell motility in the follicle.

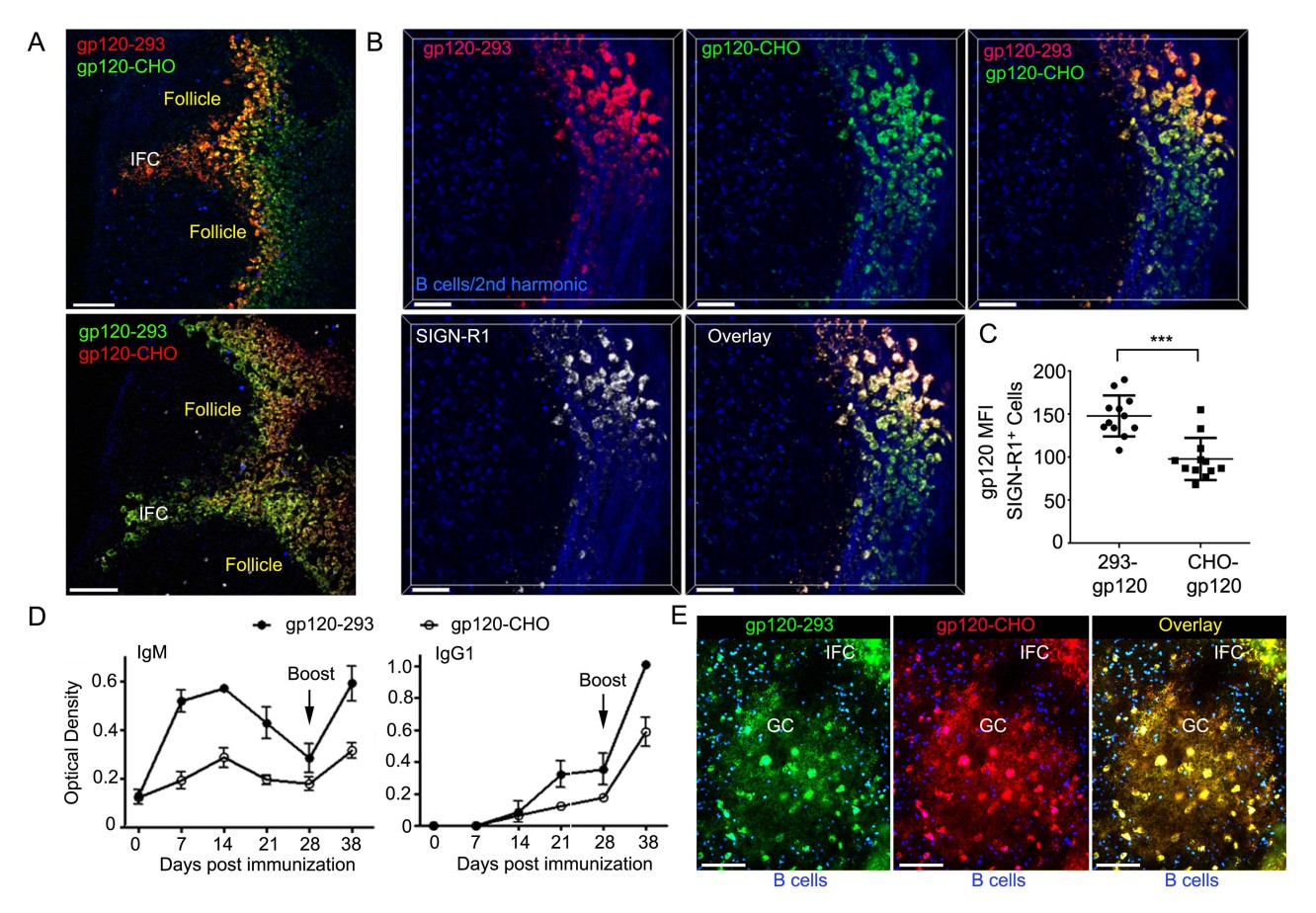

**Figure 5**. IFC network macrophages differentially uptake two different R66M gp120 preparations. (**A**) Intravital TP-LSM image of the inguinal lymph following the injection of differentially labeled R66M gp120 expressed in either 293F or CHO-S cells. Scale bars are 100 µm. (**B**) A z-projection (50 µm) of intravital TP-LSM images of the inguinal LN following the injection of differentially labeled R66M gp120 expressed in either 293F cells, red, or CHO-S cells, green, and SIGN-R1 antibody, white. Adoptively transferred B cells (blue) and the 2nd harmonic signal delineated the LN follicle and capsule. The various signals shown are indicated. Scale bars are 50 µm. (**C**) Level of gp120 binding to SIGN-R1+ cells was quantitated. The amount of 293 gp120 and CHO-gp120 bound was determined using Imaris. ***p < 0.001. (**D**) Results from ELISA assays to analyze gp120 specific antibodies present in the sera of mice at various days following immunization with R66M gp120 expressed in either 293F or CHO-S cells. Error bars, ±SEM. (**E**) A z-projection (50 µm) of intravital TP-LSM images of the inguinal LN following the local injection of R66M gp120 expressed in 293F cells, green, or CHO-S cells, red. The mice had been immunized with 293-gp120 and boosted 4 weeks later. 3 weeks after the boost the mice were injected near the inguinal LN with labeled gp120s. The day prior to the injection naïve B cells (blue) were adoptively transferred. The LN follicle images are from 4 hr after gp120 injection. Scale bar are 100 µm.

The following figure supplements are available for figure 5:

**Figure supplement 1**. Deglycosylated gp120 loses its binding specificity to SIGN-R1+ macrophages.

**Figure supplement 2**. SIGN-R1+ IFC and cortical medullary junction macrophages rapidly accumulate lymph borne soluble trimeric gp120.

## Discussion

A subset of SSMs that express SIGN-R1 captures monomeric and trimeric gp120 from the afferent lymph following its local injection. These macrophages are distinguishable from the standard subcapsular macrophages, but their localization over the IFC and by their expression of SIGN-R1 and other pattern recognition receptors. The IFC macrophages underlying these cells gradually acquire gp120. A blocking antibody to SIGN-R1 reduced the interaction of gp120 with these cells as did altering the gp120 glycan shield. Arguing that a similar subset of macrophages is present in human LN, a gp120 overlay assay revealed CD163+DC-SIGN+ cells localized near the LN follicle that bound

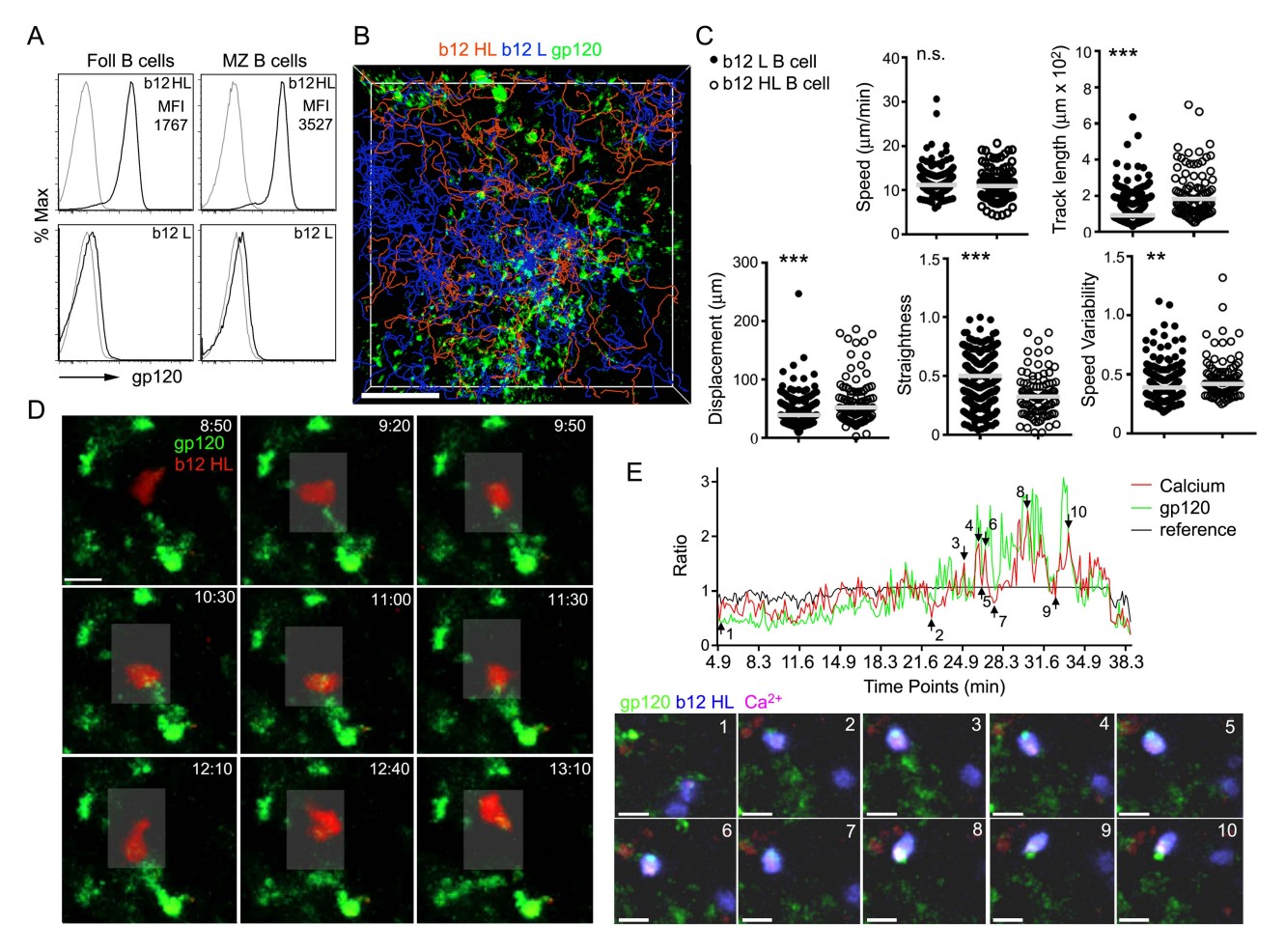

**Figure 6**. Recently arrived LN B cells that express the b12 antigen receptor can extract gp120 from IFC network cells. (**A**) Flow cytometry to evaluate the binding of labeled R66M gp120 to follicular and marginal zone B cells from either b12 HL or b12 L B cells. Light gray line is background fluorescence. MFI-mean fluorescence intensity. (**B**) Tracks of b12L and b12 HL B cells in the IFC after adoptive transfer to a mouse previously injected with fluorescent gp120. The tracks are superimposed on the IFC network delineated by gp120. Scale bar is 100 μm. (**C**) Comparison of the motility parameters generated from the analysis of b12 L and b12 HL B cells adoptively transferred into mice previously injected with gp120. Statistics calculated using unpaired t-test, **p < 0.01, ***p < 0.001. (**D**) Time laps images of a fluorescently labeled b12 HL B cell, red, approaching and departing from a cell in the IFC network that has accumulated gp120, green. Scale bar is 10 μm. Time stamps in top right, min:s. (**E**) Intravital TP-LSM imaging of b12 HL B cell labeled with eFluor 450, blue, and Calcium Orange, red, as it approaches and interacts with gp120 expressing cells, green, in the IFC. In graph, signal intensity ratio was plotted as a function of time. The ratio was calculated by dividing the intensity of an individual point by the average intensity of all the time points. Reference ratio calculated using the eFluor 450 b12 HL B cell signal. Arrows from 1 to 10 correspond to the numbered time lapse images shown below. Scale bar is 10 μm.

gp120. In the immunized mouse LN, gp120 specific B cells arriving via the HEVs could interact with, and extract gp120 from the SIGN-R1+ macrophages. Some of these macrophages extended cellular processes into the LN follicle. This allowed gp120 specific B cells located in the follicle to also acquire gp120. The B cells did not form long-lasting conjugates with the gp120 bearing IFC macrophages, but rather they repetitively, and transiently, interacted with them. B cells lacking gp120 reactive antigen receptors often wandered away showing little interest in the IFC cells. As observed previously with HEL transgenic B cells (*Suzuki et al., 2009*), the migrating gp120 specific B cells localized the extracted antigen to their uropods. These results suggest that much like germinal center B cells, which extract antigen from FDCs to acquire T cell help, naïve B cells can extract antigen from the IFC macrophages to acquire T cell help to initiate the extra-follicular antibody response and to generate germinal center precursors.

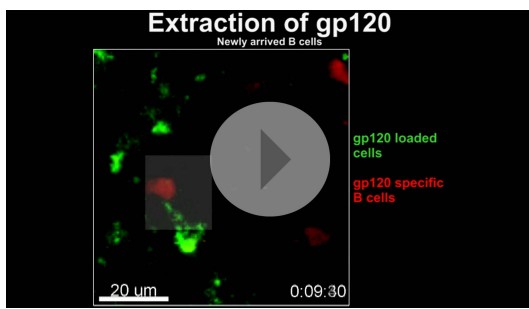

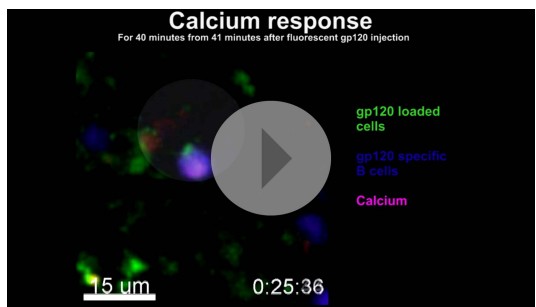

**Video 4.** Intravital TP-LSM images of a newly arriving b12 B cell extracting gp120 from IFC network cells. An image sequence of a 20 µm z-projection was acquired from the inguinal LN of a mouse, which had fluorescent gp120 (green) delineated IFC network cells. Fluorescently labeled b12 HL B cells (red) were adoptively transferred an hour after gp120 injection. Scale bar is 30 µm. Time counter shows hr:min:s.

**Video 5.** Intravital TP-LSM images of in vivo calcium response of a b12 B cell engaging a gp120 bearing IFC cell. An image sequence of a 25 µm z-projection was acquired from the inguinal LN of a mouse, which has fluorescent gp120 (green) delineated IFC network cells. b12 HL B cells (blue) were labeled with calcium orange (red). Shaded circle indicates the interaction between b12 HL B cells and gp120 loaded IFC network cells. Images were acquired for 40 min beginning 41 min after fluorescent gp120 injection into tail base. Scale bar is 15 µm. Time counter shows hr:min:s.

The IFC and cortical ridge is a crossroad for cellular traffic in the LN. Blood borne B cells enter nearby HEVs to access the LN parenchyma. The presence of the IFC macrophages and newly arrived antigen laden DCs explains why recent LN B cell entrants spend several hours exploring the IFC before entering the follicle (*Park et al., 2012*). This region likely serves as a testing ground for B cells arriving in the LN from the blood to determine whether their BCRs possess sufficient affinity to acquire antigen. The IFC macrophage antigen repertoire will reflect the specificity of their cell surface receptors for material delivered in the subcapsular sinus lymph. In the case of gp120, SIGN-R1 is functionally important for its capture by the IFC cells. However, IFC macrophages likely express other receptors that assist in gp120 acquisition since the blocking SIGN-R1 antibody only reduced the uptake of gp120 by 50%. Those B cells that do not find an IFC cell with cognate antigen migrate into the LN follicle where they remain for approximately a day (*Park et al., 2012*). During that time should new antigen arrive in the subcapsular sinus that can be captured by the IFC macrophages, cognate B cells migrating near the follicle edge can acquire it. It will be of interest to determine whether antigen loading of the IFC macrophages results in their production of EBI2 ligands, which would tend to localize LN follicle B cells toward the IFC network cells (*Hannedouche et al., 2011*; *Kelly et al., 2011*). We did note that the velocity of both the WT and the gp120 specific B cells along the follicle edge increased during the 2 hr imaging period that followed gp120 injection. The follicle B cell can also scan the FDC network for cognate antigen (*Suzuki et al., 2009*), however, until gp120 specific antibody appears little gp120 will likely be present. Those B cells that do not find cognate antigen eventually leave the follicle to exit the LN via the cortical lymphatics. Again they will have an opportunity to scan the IFC cellular network and cortical lymphatic macrophages for cognate antigen.

Although the b12 B cells carry a mutated BCR, they are functionally naïve as they have not been exposed to cognate antigen. The affinity of the b12 HL BCR for the R66M gp120 is likely low as the b12 antibody predominately recognizes B clade HIV-1 gp120 while the R66M gp120 is an A/C clade. In addition, the b12 antibody binds weakly to recombinant R66M gp120 (J Arthos, unpublished observation). As indicated above the gp120 specific B cells did not make a single long lasting synapse with the gp120 bearing IFC macrophages, but rather many transient interactions during which time the specific B cells serially extracted gp120 from the IFC cells, much like bumble bees gathering nectar. The temporal variation in gp120 fluorescence associated with the specific B cells, presumably reflects gp120-b12 HL BCR engagement, BCR internalization, gp120 degradation, and reduced fluorescence. As the B cell moves on to interact with another gp120 loaded macrophage the process is repeated. The reason why naïve B cells spend several hours repetitively engaging cognate antigen bearing cells is unclear. The serial BCR engagements and antigen extractions should provide

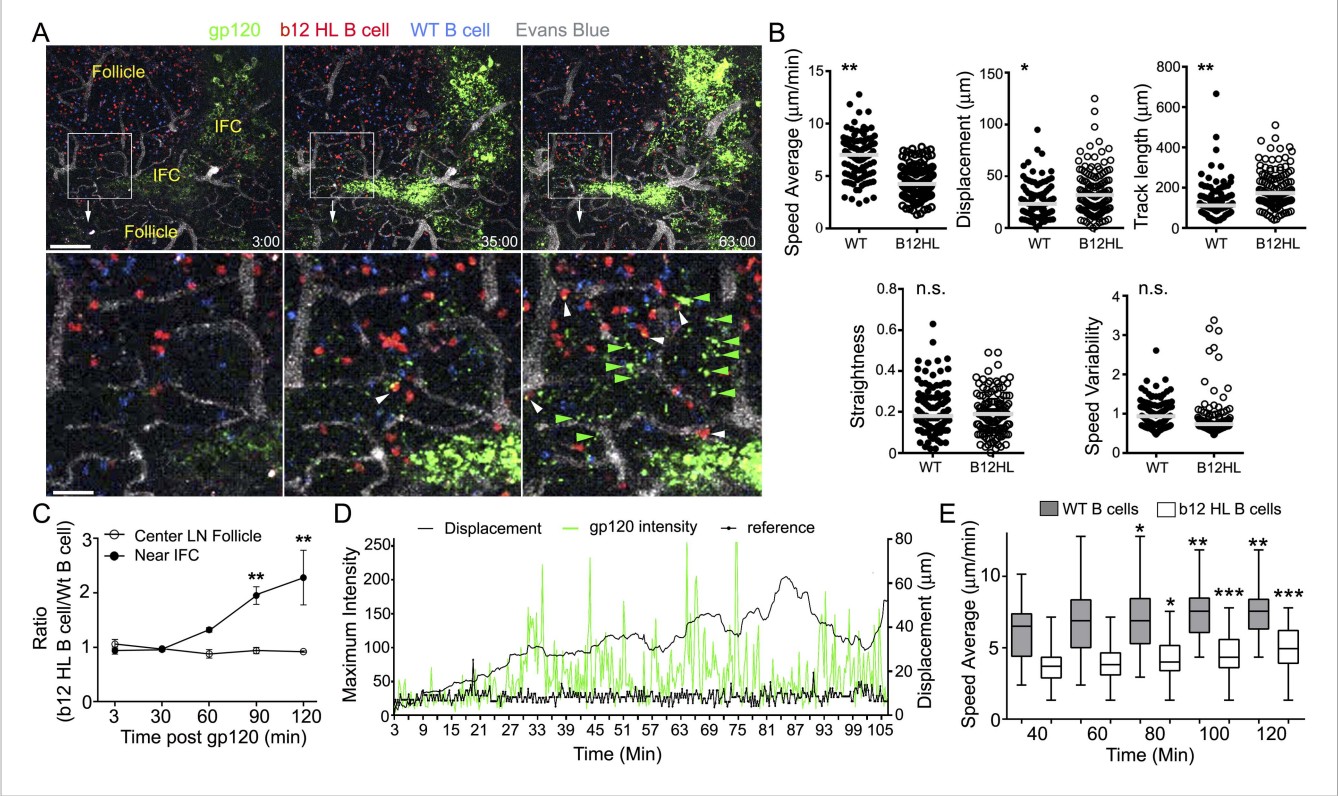

**Figure 7**. LN follicle B cells that express the b12 antigen receptor can extract gp120 from IFC network cells. (**A**) Intravital TP-LSM images of b12 HL, red, and WT B, blue, cells in the inguinal LN at 3, 35, and 63 min following injection of fluorescent gp120, green, near the inguinal LN. Blood vessels were visualized by intravenous injection of Evans blue, white. Scale bar is 100 μm. Bellow each image is an electronic zoomed image from the indicated area. White arrowheads indicate B cells that have accumulated gp120 and green arrowheads gp120 in the LN follicle. Scale bar is 25 μm. (**B**) Motility parameters. Analyses of b12 HL and wild type (WT) tracks are shown. Statistics are by unpaired t-test *p < 0.01, **p < 0.001. (**C**) The ratio between the number of WT and b12 HL B cells at various times points following antigen injection near the IFC or in the center of follicle. Error bar, ±SEM. **p < 0.002. (**D**) Tracking a b12 HL B cell located in the LN follicle following injection of gp120. Displacement and gp120 signal overlying the cell tracked over time. Graph shows the displacement, black line, from the origin and gp120 signal, green peaks, for each individual time point. The reference, thick black, is the fluorescent signal in another channel. (**E**) The average speed of b12 HL and WT B cells near the IFC channel increases over time following gp120 injection. WT and b12 HL B cells were tracked over 20 min intervals. The time shown below is the endpoint of the tracking interval. Average speed is shown as a box- and-whisker blot. The results from the later intervals were compared to the initial interval for each cell type by unpaired t-test. *p < 0.5; **p < 0.003; ***p < 0.00002.

additional B cell activation signals and saturate class II MHC molecules with peptide fragments, respectively. Those B cells that capture the highest amounts of antigen are most likely to receive help from CD4[+] T cells (*Yuseff et al., 2013*; *Avalos and Ploegh, 2014*). Our study is consistent with a recent study that showed murine B cells can very rapidly extract antigen from plasma membrane sheets decorated with antigen or an antigen surrogate (*Natkanski et al., 2013*). In that study B cells acquired antigen from APCs by invaginating and pinching off the presenting cell membrane using their BCR along with myosin IIa-mediated contractions. Since we visualized the IFC macrophages as a consequence of their gp120 binding, we could not detect whether the gp120 specific B cells also pinched off IFC macrophage membrane although the imaging suggested that might be the case as the B cells appeared to grab aggregates of fluorescent gp120. Alternatively, B cells can extract immobilized antigen by recruiting MHC class II-containing lysosomes to a B cell synapse (*Yuseff et al., 2011*). Localized lysosome exocytosis acidifies the synapse and releases hydrolases, which promote antigen extraction. In our study this scenario seems less likely due to the transient nature of the interactions.

For the gp120 specific B cells to gather gp120 from the SIGN-R1[+] macrophages, the gp120 must remain on the surface of the macrophage and not be fully internalized. SIGN-R1 functions as

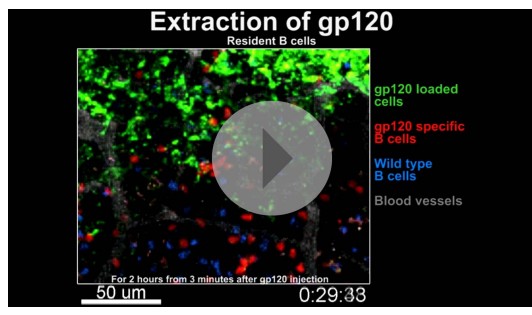

**Video 6.** Intravital TP-LSM images of resident b12 B cells that extract gp120 from the IFC network cells. An image sequence of a 20 μm z-projection was acquired from inguinal LN of mouse, which had received by adoptive transfer the previous day b12 HL B cells, red, and wild type B cells, blue. Evans blue delineated the blood vessels, gray. Fluorescent gp120, green, was injected near the tail base and the image sequence was acquired over the next 2 hr. Scale bar is 50 μm. Time counter shows hr:min:s.

a phagocytic receptor and is known to bind bacterial dextrans and the capsular polysaccharides of *Streptococcus pneumonia* (*Kang et al., 2003*, *2004*). Our data indicates that SIGN-R1 is also involved in gp120 binding and that sufficient gp120 remains surface bound for B cells to acquire it from the SIGN-R1[+] macrophages. Intravital imaging the inguinal LN of a naïve mouse 72 hr after local injection of three micrograms of labeled gp120 revealed its continued presence on IFC macrophages although the fluorescence levels had declined. This suggests that the gp120 remains available for several days following its acquisition by the IFC macrophages. Of note at the same time point gp120 was not found on the LN follicle FDCs (C Park, unpublished data). It will important to assess how long the injected gp120 remains associated with the IFC macrophages and to determine whether at later time points other cell types can acquire and provide gp120 to B cells. The eventual loading of FDCs following gp120 immunization will likely depend upon gp120 persistence, the primary antibody response, gp120 immune complex formation, the capture of immune complexes by subcapsular macrophages, and their delivery to the underlying FDC network. Consistent with that scenario within 4 hr of the injection of gp120 into previously immunized mice, we found gp120 on the FDC network; however, the IFC macrophages also continued to capture it.

The LN SIGN-R1[+] IFC macrophages preferentially captured two early viral isolate monomeric gp120 preparations and a HIV-1 envelope glycoprotein trimer that adopts a native conformation. In contrast, a monomeric gp120 protein preparation treated with Peptide-N-Glycosidase F to reduce its glycan content, the algae protein PE, or HEL showed little specificity for these cells. As many of the current recombinant vaccine candidates are being produced in HEK 293 they are also likely to be captured in the LN by the SIGN-R1[+] IFC macrophages we have described. However, the glycosylation pattern of the HIV-1 recombinant envelope proteins we tested likely differ from the HIV-1 envelope proteins produced in vivo in infected T cells and macrophages. Additional studies with gp120 or HIV virions produced in endogenously infected cells types are certainly warranted.

Disruption of the IFC cellular network by pathogens would likely limit early antibody responses to gp120 and other antigens captured by this network of macrophages. As a consequence this would reduce subsequent immune complex formation and ultimately decrease FDC network loading. The IFMs are possible HIV-1 targets as they are CD4 and CCR5 positive (*Gray and Cyster, 2012*). Furthermore, cells located within the LN IFC have been noted to be infected in HIV patients (*Schuurman et al., 1988*). Conversely, enhancing the loading and function of the IFC macrophages could improve the early humoral response to gp120 vaccines and other antigens captured by the IFC cells. Since we injected gp120 in the absence of any adjuvant, further experiments will be needed to assess how different adjuvants affect gp120 capture and its delivery to LN B cells. In addition, our study predominately focused on the events that occurred over the first several hours following gp120 injection. Many questions remain to be answered. What are the fates of the B cells that encounter antigen from the macrophages in the IFC? How does the binding of gp120 to the IFC macrophages functionally affect them? Additional studies built on the identification of this IFC cellular network should help to optimize the early extra-follicular antibody production and germinal center formation following gp120 immunization.

## Materials and methods

### Mice

C57BL/6 and C.129S4(B6)-Ifngtm3.1Lky/J (GREAT mice) were obtained from Jackson Laboratory (Bar Harbor, ME). The C57BL/6, b12 HL, b 12H, and b12 L mice were obtained from Dr David Nemazee

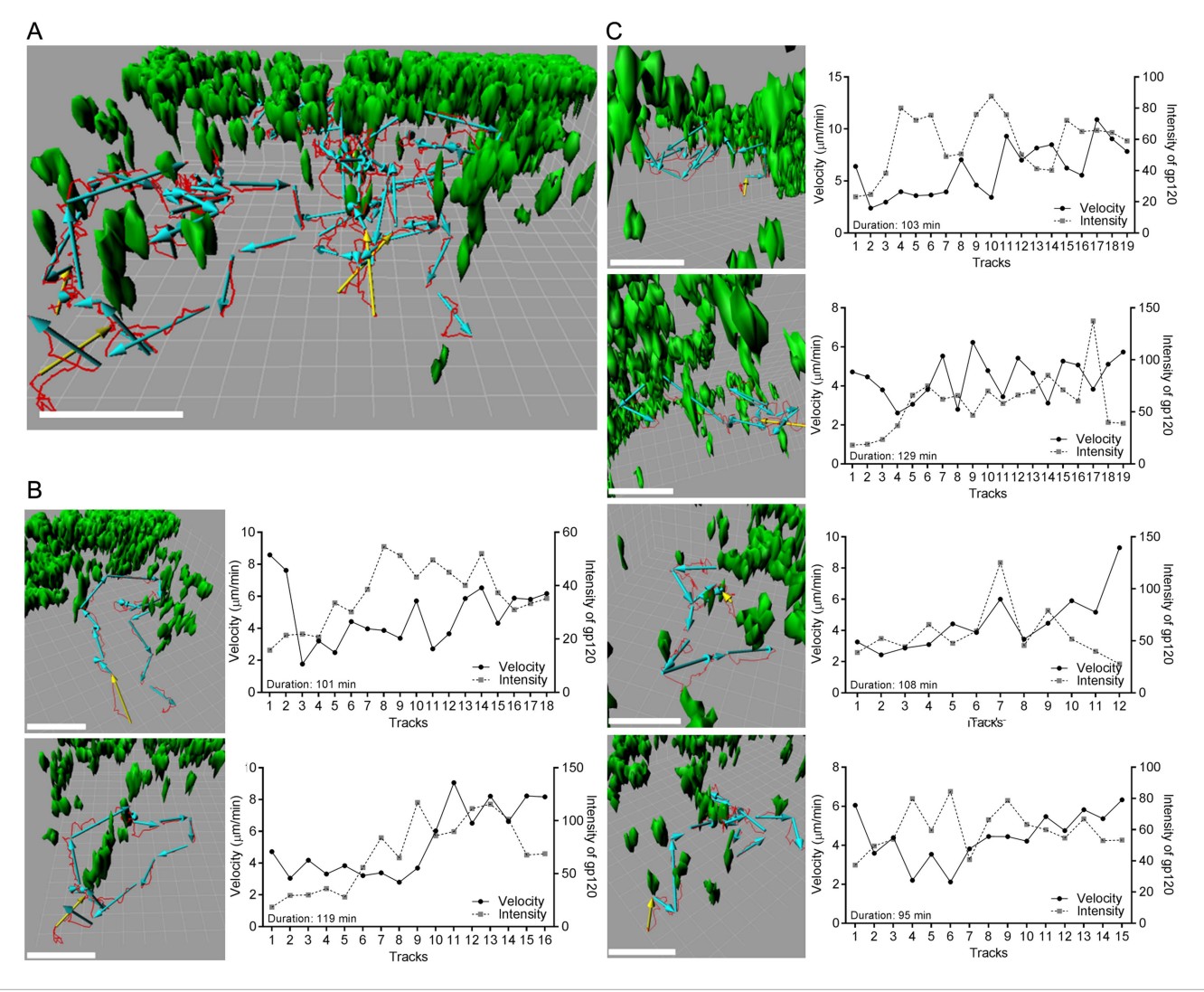

**Figure 8**. Individual b12 HL B cell tracks from b12 HL B cells located in the LN follicle near the IFC channel following injection of gp120. (**A**) Six tracks, red lines, and displacements, yellow or cyan arrows, of b12 HL B cells were superimposed on 3-D reconstruction image of gp120 loaded IFC network cells, green. 3-D reconstruction images were generated with 50 μm z-stack volume image. Displacement arrows were generated with fragments of the original track, which disconnected at time points that showed typical turning or discontinues movement from the original single track. Yellow arrows in each track indicate the starting point. Scale bar is 50 μm. Grid spacing in 3-D view is 10 μm. (**B**, **C**) Each track in (a) was visualized from a different angle to see typical tracked cell pattern (approach, survey, move away). Scale bar is 50 μm. Grid spacing in 3-D view is 10 μm. Each individual track is shown as a single track plot of velocity (solid line) and gp120 intensity (dotted line). Duration is total time span of the original single tracks in a 2 hr intravital TPLSM imaging.

and maintained at the NIH. The LysM-enhanced green fluorescent protein (EGFP) mice were kindly provided by Ron Germain (NIAID, NIH) with permission from Thomas Graf (Center for Genomic Regulation, Barcelona, Spain). All mice were used in this study were 6–14 weeks of age. Mice were housed under specific-pathogen-free conditions. All the animal experiments and protocols used in the study were approved by the NIAID Animal Care and Use Committee (ACUC) at the National Institutes of Health.

## Recombinant gp120

The coding sequences of the R66M (A/C R66M 7Mar06 3A9env2) envelope protein, from +1 to the gp120-gp41 junction was inserted into a mammalian expression vector downstream of a synthetic leader sequence. The coding sequence was provided by Dr Cynthia Derdeyn (Emory University).

The vector was transiently transfected into either 293F or CHO-S cells using FreeStyle MAX Reagent (Invitrogen, Thermo Fisher Scientific, Waltham, MA) per the manufacturer's instructions. Protein-containing supernatants were harvested 72 hr after transfection and were passed over a column of lectin sepharose from *Galanthus nivalis* (Vector Laboratories, Burlingame, CA), which was diluted 1:5 with sepharose 4B not bound to ligand to minimize avid binding. gp120 was eluted with 20 mM glycine-HCl, pH 3.0, 150 mM NaCl, 500 mM α-methyl-mannopyranoside (Sigma Aldrich, St. Louis, MO), in 5-ml fractions, directly into 1 M Tris-HCl, pH 8.0. Peak fractions were pooled, concentrated with a stirred cell concentrator (EMD Millipore, Bilerica, MA) and dialyzed exhaustively against HEPES, pH 7.4, 150 mM NaCl. Higher molecular weight forms were removed by size-exclusion chromatography. To eliminate possible endotoxin contamination from purified proteins, a Triton X114 extraction was done. Proteins were quantified by ultraviolet absorption at a wavelength of 280 nm (extinction coefficient, 1.1) and values were confirmed by a bicinchoninic acid protein assay (Thermo Fisher Scientific). Biotinylated soluble HIV-1 Env trimeric gp120, BG505 SOSIP, was kindly provided by Dr John P Moore (*Sanders et al., 2013*).

## Fluorescent conjugation of gp120, antigen, antibodies and infusion

Recombinant gp120 and HEL were conjugated to fluorescent (Alexa Fluor 488, 594, or 647) with the Microscale Protein Labeling Kit (Molecular probe, Thermo Fisher Scientific). Antibodies against CD169 (3D6.112, BioLegend, San Diego, CA), SIGN-R1 (eBio22D1, eBiosciences, San Diego, CA), LYVE-1 (Clone# 223322, R&D System, Minneapolis, MN) and ER-TR7 (ER-TR7, AbD serotec, Bio-Rad Laboratories, Hercules, CA) were conjugated to Alexa Fluor 488, 594, or 647 with the Antibody Labeling Kits (Molecular probe). Labeling reactions, conjugates purification, and determination of degree of labeling were performed following the company manuals. 6–10-week-old recipient, anesthetized mice were injected with fluorescent labeled materials for intravital imaging or section imaging by tail base injection.

## Intravital TP-LSM

Inguinal LNs were prepared for intravital microscopy as described (*Park et al., 2009*, *2012*). Cell populations were labeled for 10 min at 37°C with 2.5–5 µM red cell tracker CMTMR (Molecular probes) or 2 µM of eFluor450 (eBioscience). 5–10 million labeled cells of each population in 200 ml of PBS were adoptively transferred by tail vein injection into recipient mice. After anesthesia the skin and fatty tissue over inguinal LN were removed. The mouse was placed in a pre-warmed coverglass chamber slide (Nalgene, Nunc, Thermo Fisher Scientific). The chamber slide was then placed into the temperature control chamber on the microscope. The temperature of air was monitored and maintained at 37.0 ± 0.5°C. Inguinal LN was intravitally imaged from the capsule over a range of depths (10–220 µm). All two-photon imaging was performed with a Leica SP5 inverted 5 channel confocal microscope (Leica Microsystems, Wetzlar, Germany) equipped with 25× water dipping objective, 0.95 NA (immersion medium used distilled water). Two-photon excitation was provided by a Mai Tai Ti:Sapphire laser (Spectra Physics, Newport Research Corporation, Invine, CA) with a 10 W pump, tuned wavelength ranges from 810 to 910 nm. Emitted fluorescence was collected using a 4 channel non-descanned detector. Wavelength separation was through a dichroic mirror at 560 nm and then separated again through a dichroic mirror at 495 nm followed by 525/50 emission filter for GFP or Alexa Fluor 488 (Molecular probes); and the eFluor450 (eBioscience) or second harmonic signal was collected by 460/50 nm emission filter; a dichroic mirror at 650 nm followed by 610/60 nm emission filter for CMTMR, PE or Alexa Fluor 594; and the Evans blue or Alexa Fluor 647 signal was collected by 680/50 nm emission filter. For four-dimensional analysis of cell behavior, stacks of various numbers of section (z step = 3, 4, or 6 µm) were acquired every 10–12 s to provide an imaging volume of 30–100 µm in depth. Sequences of image stacks were transformed into volume-rendered four-dimensional videos using Imaris software v.7.7.1 64× (Bitplane AG, Zurich, Switzerland), and the tracks analysis was used for semi-automated tracking of cell motility in three dimensions by using the following parameters: autoregressive motion algorithm, estimated diameter 10 µm, background subtraction true, maximum distance 20 µm, and maximum gap size 3. Tracks acquired that could be tracked for at least 20% of total imaging duration were used for analysis. Some tracks were manually examined and verified. Calculations of the cell motility parameters (speed, track length, displacement, straightness and speed variability) were performed using the Imaris software v.7.7.1 64× (Bitplane AG). Statistical

analysis was performed using Prism software (GraphPad Software, La Jolla, CA). 3D-reconstructions from original images from TP-LSM were generated by the surfaces function of the Surpass view in Imaris software v.7.7.1 64× (Bitplane AG), performed with semi-automated creation wizard. Annotations on videos and video editing were performed using Adobe Premiere Pro CS3 (Adobe Systems Incorporated, McLean, VA). Video files were converted to MPEG4 format with Imtoo Video Converter Ultimate 6.0.2 for Mac (Imtoo Software Studio).

## Immunohistochemistry and confocal microscopy

Immunohistochemistry was performed using a modified method of a previously published protocol (Chai et al., 2013). Briefly, freshly isolated LNs or spleens were fixed in newly prepared 4% paraformaldehyde (Electron Microscopy Science, Hatfield, PA) overnight at 4°C on an agitation stage. Spleens or LNs were embedded in 4% low melting agarose (Invitrogen) in PBS and sectioned with a vibratome (Leica VT-1000 S) at a 30 μm thickness. Thick sections were blocked in PBS containing 10% fetal calf serum, 1 mg/ml anti-Fcγ receptor (BD Biosciences), and 0.1% Triton X-100 (Sigma Aldrich) for 30 min at room temperature. Sections were stained overnight at 4°C on an agitation stage with the following antibodies: anti-B220 (RA3-6B2, BD Biosciences), anti-CD3e (17A2, BD Biosciences), anti-CD4 (RM4-5, BD Biosciences), anti-CD11c (HL3, BD Biosciences), anti-CD169 (3D6.112, BioLegend), anti-ER-TR7 (ER-TR7, AbD serotec) and anti-CD21/35 (BioLegend) and with labeled gp120. For the human LN analysis, cold acetone fixed frozen sections (cat#. T1234161) were purchased from BioChain Institute, Inc, (Newark, CA). Sections were fixed again with 4% paraformaldehyde for 10 min at room temperature. Then sections were blocked in PBS containing 10% fetal calf serum, 1 mg/ml human IgG (Sigma Aldrich), and 0.1% Triton X-100 (Sigma Aldrich) for 30 min at room temperature. Sections were stained overnight at 4°C on an agitation stage with the following antibodies: anti-CD19 (4G7, BD Biosciences), anti-CD4 (RPA-T4, eBiosciences), anti-DC-SIGN (DCN46, BD Biosciences), anti-CD163 (eBioGHI/61, eBiosciences) and anti-CD11c (S-HCL-3, BD Biosciences) or with gp120. For the trimeric gp120, biotinylated trimeric gp120 was injected into tail base for 2 hr and detected by AlexaFluor 488 conjugated streptavidin in LN sections. Stained thick sections and human LN sections were microscopically analyzed using a Leica SP5 confocal microscope (Leica Microsystem, Inc.) and images were processed with Leica LAS AF software (Leica Microsystem, Inc.) and Imaris software v.7.7.1 64× (Bitplane AG).

## Deglycosylation of gp120

Recombinant of gp120 was deglycosylated with Peptide-N-Glycosidase F (PNGase F, New England Biolabs, Ipswich, MA). In order to minimize the denaturation of gp120, recombinant gp120 was deglycosylated with a company protocol that uses non-denaturing reaction conditions. One μg of fluorescent gp120 (AlexaFluor 647 conjugated) was incubated at 37°C for 20 hr in reaction mixture of 1 unit of PNGase F, 2 μl of 10X GlycoBuffer and $dH_2O$ to make a 20 μl total reaction volume. In determine whether the degylcosylation affected the fluorescent signal, the deglycosylated gp120 and reaction control (without PNGase F) were analyzed on a 10% NuPAGE Bis-Tris gel (Life technologies, Thermo Fisher Scientific), and the strength of fluorochrome was measured by Odyssey CLx Infrared Imaging System (LI-COR, Inc., Lincoln, NE).

## Visualization of microvessels included HEVs and lymphatics in the LN

To outline blood vessels 50 μl of Evans Blue solution (0.5 μg/ml in PBS) was injected into orbital or tail vein prior to imaging. HEVs were delineated via the presence of adherent T-cells previously adoptively transferred into the tail vein. To visualize endothelial cells in the lymphatic sinuses, purified rat anti-mouse LYVE-1 was conjugated with Alexa Fluor 647. Five μg of AlexFluor 647 conjugated LYVE-1 antibody in 50 μl of PBS was injected into tail base 1 hr prior to imaging.

## Imaging in vivo $Ca^{2+}$ responses

To visualize changes in intracellular calcium in b12 HL B cells, the cells were loaded with Calcium Orange (Invitrogen). A single cell suspension of b12 HL B cells in culture media was incubated with 5 μM of Calcium Orange (5 mM stock solution in Fluronic F-127 [20% [wt/vol]solution in DMSO]), at room temperature for 30 min. The stained cells were washed five times with culture media prior to being adoptively transferred. The Calcium Orange signal was imaged by TP-LSM.

## Quantitation of fluorescent signals

The intensities of fluorescent signals in ROIs were measured by LSA AF Lite software (Leica Microsystem, Inc.). To make the relative mean intensity score, Alexa Fluor 647 conjugated gp120 intensity from SIGN-R1$^+$ sinus macrophages (4 cells) was divided by signal intensity from sinus lumen (1 area) as a reference signal. The gp120 intensity from the network cells (4 cells) was determined in the same manner. The generated mean intensity score was plotted as a function of time. The intensity scores of gp120 and NP-PE were calculated using intensity of HEL as a reference signals in ROIs. The intensity of fluorescent signals in individual tracked cells was measured by Imaris software v.7.7.1 64× (Bitplane AG). The cell volume was reconstructed by the surface function in Imaris software, tracked, and evaluated manually. The intensity ratio of each signal at the indicated time points was calculated using mean intensity of each signal over the entire imaging period.

## Preparation of LN cells, flow cytometry, and FACS sorting

Inguinal LNs were carefully collected without fat tissue and gently teased apart with micro-forceps into RPMI 1640 media containing 2 mM L-glutamine, antibiotics (100 IU/ml penicillin, 100 µg/ml streptomycin), 1 mM sodium pyruvate, and 50 µM 2-mercaptoethanol, pH 7.2. The tissue was then digested with Liberase Blendzyme 2 (0.2 mg/ml, Roche Applied Science, Penzberg, Germany) and DNase I (20 µg/ml) for 30 min at 37°C, while rocking vigorously. Proteases were then inactivated with 10% fetal bovine serum and 2 mM EDTA and the cell disaggregated by passing them through a 40 µm nylon sieve (BD Bioscience). Single cells were then washed with 1% BSA/PBS and blocked with anti-Fcγ receptor (BD Biosciences). LIVE/DEAD Fixable Aqua Dead Cell Stain Kit (Molecular Probes) was used in all experiments to exclude dead cells. Single cells were re-suspended in PBS, 2% FBS, and stained with fluorochrome-conjugated or biotinylated antibodies against Gr-1 (RB6-8C5, eBioscience or BD Bioscience), CD169 (3D6.112, BioLegend), SIGN-R1 (eBio22D1, eBiosciences), anti-B220 (RA3-6B2, BD Bioscience), anti-CD4 (clone RM4-5, BD Bioscience), anti-CD11b (M1/70, eBiosciences), anti-CD11c (HL3, BD Bioscience), anti-F4/80 (BM8, eBiosciences). Data acquisition was done on FACSCanto II (BD Bioscience) flow cytometer and analyzed with FlowJo software (FLOWJO, LLC, Ashland, OR). FACS Sorting of gp120 positive cells was performed with LN cells prepared as was done for flow cytometry. The suspended LN cells were immunostained and applied to a FACS-Aria, which was set for 5–6 droplets through 100 µm nozzle (20 psi). Sorted cells were directly visualized with confocal microscope and cultured with standard media containing 20 ng/ml of M-CSF for 7 days before analysis.

## In vitro binding of gp120 to LN cells

The same fluorescently labeled gp120 used in the in vivo experiments was used in the in vitro binding assay. Single cells suspensions were prepared, washed with 1% BSA/PBS, and blocked with anti-Fcγ receptor (BD Biosciences). LIVE/DEAD Fixable Aqua Dead Cell Stain Kit (Molecular Probes) was used to exclude dead cells. Single cells were re-suspended in PBS, 2% FBS, and stained on ice with various antibodies and fluorescent gp120 (1 µg). In some instances non-labeled gp120 (2 µg) or mouse SIGN-R1/CD209b antibody (2 µg, R&D System) was added for 30 min prior to immunostaining and the addition of fluorescent gp120. Data acquisition was done on FACSCanto II (BD Bioscience) flow cytometer and analyzed with FlowJo software (FLOWJO, LLC).

## Intracellular flow cytometry to detect interferon-γ and interferon-γ- eYFP reporter mice

3 hr after gp120 injection the draining inguinal LNs were collected and prepared for flow cytometry. The cells were immunostained using the BD Cytofix/Cytoperm Fixation/Permeabilization Kit protocol (BD Bioscience). Briefly single cell suspensions were fixed and permeabilized with Fixation/Permeabilization solution for 20 min at 4°C. The cells were immunostained with PE-Cy7 conjugated interferon-γ antibody (1:100 dilution of XMG1.2, eBiosciences) overnight. The cells were then pelleted, washed, and resuspended with 1× BD Perm/Wash buffer. All flow cytometry data were collected on a BD FACS CANTO II and analyzed with FlowJo software (FLOWJO, LLC). The level of eYFP expression in C.129S4(B6)-Ifngtm3.1Lky/J (GREAT mice) after gp120 injection was measured with LN cell suspension prepared as outlined for flow cytometry analysis.

## Immunization and ELISA

Each group of C57Bl/6 mice was immunized with gp120 prepared from either 293F or CHO-S cells. The recombinant gp120 (50 µg) was mixed with Imject Alum (Thermo Fisher Scientific) and injected subcutaneously. Mice were boosted with same dose of antigen at the indicated days along with Alum. Serum gp120 specific Ig levels in these mice were analyzed by ELISA. Briefly, 96 well ELISA plates (Nalgene, Nunc) were coated with gp120 (0.8 µg/well) overnight at 4°C, washed and blocked with 1% BSA fraction V (Sigma–Aldrich), serum titers were then added to the plates and incubated 4 hr at 4°C. After washing alkaline phosphatase-labeled goat anti-mouse IgM or IgG isotype specific antibodies were added for 2 hr at room temperature (SouthernBiotech, Birmingham, AL). After washing, PNPP one component substrate (SouthernBiotech) was used to detect the amount of antibody bound.

## Statistics

All experiments were performed at least three times. Primary data calculated by Imaris (Bitplane AG) was acquired and processed with Microsoft Excel software. Error bars with $\pm$SEM, and p values were calculated with GraphPad Prism (GraphPad software) as a function of linear regression in XY analyses (slope, *Figure 2B*), 2way ANOVA (*Figures 2B, 4E*), unpaired t-test (*Figure 7C*).

## Acknowledgements

The authors are grateful to Dr David Nemazee for providing the b12 KI mice, Drs John P Moore, Rogier W Sanders, and Marit J van Gils for providing the soluble trimeric gp120 and would like to thank Dr Anthony Fauci for his continued support. This research was supported by the intramural program of the National Institutes of Allergy and Infectious Diseases.

## Additional information

### Funding

| Funder | Author |
| --- | --- |
| Division of Intramural Research, National Institute of Allergy and Infectious Diseases (Division of Intramural Research of the NIAID) | Chung Park, James Arthos, Claudia Cicala, John H Kehrl |

The funder had no role in study design, data collection and interpretation, or the decision to submit the work for publication.

### Author contributions

CP, Conception and design, Acquisition of data, Analysis and interpretation of data, Drafting or revising the article; JA, Analysis and interpretation of data, Contributed unpublished essential data or reagents; CC, Conception and design, Analysis and interpretation of data; JHK, Conception and design, Analysis and interpretation of data, Drafting or revising the article

### Ethics

Animal experimentation: This study was performed in strict accordance with the recommendations in the Guide for the Care and Use of Laboratory Animals of the National Institutes of Health. All of the animals were handled according to approved institutional animal care and use committee (IACUC) protocols (LIR-15) of the National Institute of Allergy and Infectious Diseases.

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
