## [Decision Letter]

Thank you for sending your work entitled “Capture of HIV gp120 by SIGN-R1
positive lymph node macrophages provides a platform for gp120 specific B cell
activation” for consideration at *eLife*. Your article has been
favorably evaluated by Tadatsugu Taniguchi (Senior Editor), a Reviewing Editor, and two
reviewers.

The Reviewing Editor and the reviewers discussed their comments before reaching this
decision, and the Reviewing Editor has assembled the following comments to help you
prepare a revised submission.

To understand how antibody responses are mounted against HIV gp120 it is important to
define how the antigen is captured and displayed for B cell recognition. The current
manuscript by Kehrl and coworkers addresses this issue using the mouse as a model. They
find gp120 is preferentially captured by cells in interfollicular regions of draining
LNs, including SIGN-R1^+^ macrophages. This is a distinct pattern of
capture from that reported for immune complexes, antigen-coated beads and VSV, which
were all found in previous studies to be preferentially bound and displayed by
CD169^+^ subcapsular sinus macrophages (SSMs). The gp120 binding
pattern is most similar to that previously reported for inactivated influenza virus
(9) and although the
authors suggest there is little similarity to that pattern, the central conclusion of
the influenza study, like the present one, was that SIGN-R1^+^LN
macrophages were involved in antigen capture and presentation to B cells. The focus on
interfollicular regions in the present work is considered a positive feature of the
study, though it should be noted that the distinction between interfollicular and
medullary regions is not always straightforward as B cells follicles can be proximal to
the medulla.

In a series of well performed experiments, this study describes the time course of
labeled gp120 appearance on interfollicular cells, and shows this distribution differs
from that of a small soluble antigen (HEL) that accesses LN conduits. Cognate B cells
are shown to encounter and capture gp120 from interfollicular cells, to transmit
Ca^2+^ signals during this process, and to transiently modulate their
migration during interactions leading to antigen capture. They also find that a form of
gp120 with higher sialylation gains deeper access to interfollicular regions and this is
correlated with a more robust antibody response.

Overall, this study provides important new information about the distinct ways in which
a highly glycosylated antigen is handled in the LN during a primary immune response,
describing properties of the cell types involved and demonstrating that cognate B cells
can capture antigen from these cells. Perhaps the one significant concern I have is
whether the antigen-capture and display logic observed for gp120 in the mouse applies in
humans. Since HIV1 is not a natural pathogen of mice, to understand how directly
translatable these findings are for gp120 antigen handling in humans it would seem
reasonable to ask if any aspect of the antigen handling can be recapitulated with human
or primate tissue. In particular, the fluorescent gp120 overlay experiment of the type
used in Figure 2 could be attempted with human or
primate LN sections.

Other comments:

1) Published work has highlighted the difficulty of isolating LN macrophages in pure
form (e.g. Gray et al., 2012 PLoS One 7, e38258). To establish that the cells analyzed
in Figure 1 are macrophages and to
validate the statement that the gp120 binding cells are
SIGN-R1^+^/CD169^mid^/CD11b^mid^/CD4^+^/
CD11c^-^/F4/80^low^ sinus and interfollicular macrophages
(subsection “Locally injected gp120 is captured in the LN by
SIGN-R1^+^ subcapsular macrophages and SIGN-R1^+^ IFC
macrophages”), microscopy analysis of isolated cells should be provided. The
IFNγ data in Figure 1 are not especially
convincing as the shift is small and it is not stated how many experiments are
represented by these data. The conclusion might be stronger if it were backed up by a
second measurement such as using the IFNγ-eYFP reporter mice available live from
Jax.

2) The data showing the SIGN-R1 dependence of gp120 binding (Figure 1) are not especially convincing. It is not clear how many
experiments are represented by the data shown or what is the fold difference in binding.
This result might be stronger if the effect of SIGN-R1 blocking in LN sections on the
ability of gp120 to bind were shown. Alternatively, data could be provided for cells
from SIGN-R1 KO mice. If the binding of gp120 really were selectively dependent on
SIGN-R1 then the authors would be in a position to perform a mechanistic test of whether
antigen capture by SIGN-R1^+^ cells is important for induction of
anti-gp120 antibody responses by immunizing SIGNR1 deficient or blocked mice.

3) The authors show, using gp120, that it is deposited on SIGN-R1 positive macrophages.
They then used two unglycosylated proteins, HEL and PE, which were not deposited on the
cells, and on the basis of this, they have concluded that glycosylation on gp120 causes
the retention. This needs to be proved more convincingly, for example by treating gp120
with glycosylases to remove glycosylation from GP120 itself and showing that the treated
protein is no longer retained. Similarly, can a SIGN-R1 blocking antibody inhibit
antigen retention? As its stands this is only shown in vitro which could lead to
artifacts.

Another possibility would be to use gp120 produced in cells treated with glycosylase
inhibitors such as Kifenensen.

4) To quantify B cell response to gp120 in the presence of inhibitory SIGN-R1 antibody.
Is the deposition on these macrophages the only way B cells do see gp120?

5) It is well established that SIGN-R1 is expressed on medullary macrophages and
dendritic cells but not in interfollicular regions. A key finding of this work is that
SIGN-R1^+^ macrophages also appear to be present in interfollicular
regions. It would be important to have a more definitive examination of this. Whole
lymph node imaging providing a comparison between medullary SIGN-R1 staining and the
described SIGN-R1 staining in interfollicular regions would significantly add to this
work. Such an analysis would also benefit from the addition of a CD11c marker. As
recently demonstrated by Ron Germain's group, dendritic cells in interfollicular
regions (LS DCs) are able to capture antigen; it would therefore be important to
demonstrate that SIGN-R1 interfollicular macrophages are actually capturing the gp120
and exclude the possibility that this is coming from DCs.

6) The authors show at the beginning of the manuscript that gp120 co-localizes with
SIGN-R1. From then onwards, they mostly rely on only the localization of gp120 and not
SIGN-R1 as well, which may lead to confusion. It is important to have this control in
all the figures.

7) Is this the case for whole of HIV? It would be great if the authors were to consider
the injection of inactivated HIV. However, something that the authors will have to deal
with is to discuss their findings in the context of the wider literature. It is
important to recognize that several groups, such as the one of Uli Von Adrian, have been
making seminal contributions on virus retention in the follicles. Similarly, the
behavior of the B cells looks similar to the one described during antigen extraction
from FDCs or in vitro, which has been reported by both the Cyster group and by the
Neuberger group, the latter having actually introduced the term.

*Should you be unable to use* HIV for these experiments you can use a
more native trimer preparation such as the ones that have been crystalized by Ian Wilson
and Peter Kwong consisting of BG505 native trimers.

[Editors' note: further revisions were requested prior to acceptance, as
described below.]

Thank you for resubmitting your work entitled “The HIV-1 envelope protein gp120
is captured and displayed for B cell recognition by SIGN-R1^+^ lymph
node macrophages” for further consideration at *eLife*. Your
revised article has been favorably evaluated by Tadatsugu Taniguchi (Senior Editor), a
Reviewing Editor, and two reviewers. The manuscript has been improved but there are some
remaining issues that need to be addressed, as outlined below:

1) Data are added to show SIGN-R1^+^ staining of isolated LN macrophages
by fluorescence microscopy of sorted cells (Figure 1—figure supplement 1). However, in the example shown it appears that 1
cell out of ∼4 is SIGNR1^+^. It needs to be indicated what
frequency of the sorted cells have a macrophage appearance and SIGNR1^+^
surface staining.

2) Related to 1, the amount of IFNγ-YFP expression by the
‘macrophages’ is surprising given the limited evidence in the literature
for IFNγ expression by macrophages. Without knowing how well excluded T cells, NK
cells and ILCs are from the macrophage gate (e.g. by frequency determination for several
experiments in 1 above), it is difficult to feel confident about the IFNγ
data.

3) Given that the binding to SIGN-R1^+^ cells strongly depends on
glycosylation, how well matched is the glycosylation of gp120 produced in CHO-S cells or
293F cells to the form produced in vivo by infected T cells and macrophages? The
cellular tropism of highly glycosylated proteins may reflect more the cell types in
which they are expressed than unique properties of the antigen. In this regard, it is
notable that the “two other antigens” mentioned in the Abstract as not
being captured by SIGN-R1^+^ macrophages are also not made in CHO-S or
293F cells and neither one is a glycoprotein. Some text should be added to the
Discussion to address this issue (perhaps commenting that the antigen studied may be
most relevant to vaccine type antigens) and to highlight the need for future studies
with gp120 or HIV virions produced from endogenously infected cell types.

---

## [Author Response]

*[…] Overall, this study provides important new information about the
distinct ways in which a highly glycosylated antigen is handled in the LN during a
primary immune response, describing properties of the cell types involved and
demonstrating that cognate B cells can capture antigen from these cells. Perhaps the
one significant concern I have is whether the antigen-capture and display logic
observed for gp120 in the mouse applies in humans. Since HIV1 is not a natural
pathogen of mice, to understand how directly translatable these findings are for
gp120 antigen handling in humans it would seem reasonable to ask if any aspect of the
antigen handling can be recapitulated with human or primate tissue. In particular,
the fluorescent gp120 overlay experiment of the type used in*
Figure 2
*could be attempted with human or primate LN sections*.

We visualized DC-SIGN^+^/CD163^+^ macrophages in a human
lymph node section with a fluorescent gp120 overlay assay. This data is shown in an
added supplemental figure (Figure 2—figure supplement 2). These cells are similar to the SIGN-R1^+^
macrophages in mouse lymph node and they are located in an area adjacent to the lymph
node follicle in a lymphatic sinus rich area. CD163^+^ is a human
macrophage marker (J Pathol., 2006 208:574-89).

*Other comments*:

*1) Published work has highlighted the difficulty of isolating LN macrophages in
pure form (e.g. Gray et al., 2012 PLoS One 7, e38258). To establish that the cells
analyzed in*
Figure 1
*are macrophages and to validate the statement that the gp120 binding cells are
SIGN-R1*^*+*^*/CD169*^*mid*^*/CD11b*^*mid*^*/CD4*^*+*^*/
CD11c*^*-*^*/F4/80*^*low*^
*sinus and interfollicular macrophages (subsection “Locally injected gp120
is captured in the LN by SIGN-R1*^*+*^
*subcapsular macrophages and SIGN-R1*^*+*^
*IFC macrophages”), microscopy analysis of isolated cells should be
provided*.

We agree that isolating LN macrophages is problematic as they are often associated with
other cell types. Our initially sorted cells were predominately macrophages based on
their behavior and morphology. To eliminate other cell types we cultured the sorted cell
with in M-CSF containing media for 7 days. The majority of these cultured cells captured
gp120 in an overlay assay, and all the capturing cells were SIGN-R1^+^.
This data is shown in Figure 1—figure supplement 1.

*The IFNγ data in*
Figure 1
*are not especially convincing as the shift is small and it is not stated how
many experiments are represented by these data. The conclusion might be stronger if
it were backed up by a second measurement such as using the IFNγ-eYFP reporter
mice available live from Jax*.

To provide more convincing data we made use of the INF-γ YFP reporter mouse. We
immunized these mice with gp120 and used a FACS assay to assess YFP expression. These
results were very similar to what we had observed previously using the intracellular
FACS with the interferon-γ specific antibody. This data is shown in Figure 1—figure supplement 2.

*2) The data showing the SIGN-R1 dependence of gp120 binding (*Figure 1*) are not
especially convincing. It is not clear how many experiments are represented by the
data shown or what is the fold difference in binding*.

We performed this experiment three times. Each of the experiments gave similar results
and one such experiment is shown in Figure 1. We
observed an inhibitory effect of pre-incubating the cells with a SIGN-R1 blocking
antibody. The % of gp120^+^ cells in gate b of Figure 1 was reduced from 41.8 to 23.7 percent. We have noted this
in the text. When we injected the SIGN-R1 blocking antibody in vivo, we also saw a
similar reduction in the percent of gp120^+^ cells in gate b (data not
shown). While our data indicates that SIGN-R1 is important for binding gp120, other
receptors likely help capture it.

*This result might be stronger if the effect of SIGN-R1 blocking in LN sections
on the ability of gp120 to bind were shown*.

We tried pre-incubating the sections with the SIGN-R1 antibody prior to the gp120
overlay. While we saw some reduction in gp120 binding, it was difficult to quantitate
the results due to the intra-assay differences in gp120 binding in the absence of
antibody. As such, we decided not to show this data.

*Alternatively, data could be provided for cells from SIGN-R1 KO
mice*.

We have tried to acquire the SIGN-R1 KO mice without success. We are still pursuing
acquiring these mice.

*If the binding of gp120 really were selectively dependent on SIGN-R1 then the
authors would be in a position to perform a mechanistic test of whether antigen
capture by SIGN-R1*^*+*^
*cells is important for induction of anti-gp120 antibody responses by immunizing
SIGN-R1 deficient or blocked mice*.

As stated above we do not think that SIGN-R1 is the only receptor capable of binding
gp120. However, we agree with the reviewer that looking at the early gp120 antibody
response in SIGN-R1 KO mice would be very interesting (see above). In addition, we think
that SIGN-R1 is very good marker for these macrophages.

*3) The authors show, using gp120, that it is deposited on SIGN-R1 positive
macrophages. They then used two unglycosylated proteins, HEL and PE, which were not
deposited on the cells, and on the basis of this, they have concluded that
glycosylation on gp120 causes the retention. This needs to be proved more
convincingly, for example by treating gp120 with glycosylases to remove glycosylation
from GP120 itself and showing that the treated protein is no longer retained.
Similarly, can a SIGN-R1 blocking antibody inhibit antigen retention? As its stands
this is only shown in vitro which could lead to artifacts*.

To address this issue we treated gp120 with PNGaseF, an amidase that cleaves between the
innermost GlcNAc and asparagine residues of high mannose, hybrid, and complex
oligosaccharides from *N*-linked glycoproteins. The native gp120 and the
PNGaseF treated gp120 were used in a mouse lymph node overlay assay. As usual the native
gp120 bound strongly to the SIGN-R1 positive macrophages, while the PNGaseF treated
gp120 exhibited much less selectivity binding to many different cell types. This
resulted in a reduction in the co-localization between SIGN-R1 and gp120. This data is
shown in Figure 5—figure supplement 1.
The injection of the SIGN-R1 antibody (in vivo) reduced the amount of binding to the
SIGN-R1 positive macrophages by a similar amount to that noted in the in vitro assays
(data not shown).

*Another possibility would be to use gp120 produced in cells treated with
glycosylase inhibitors such as Kifenensen*.

We agree that this would be useful experiment. We are in the process of obtaining gp120
produced in cells treated with Kifunensine, however, we do not yet have the protein. The
Kifunensine treated cells should produce a high mannose gp120.

4) To quantify B cell response to gp120 in the presence of inhibitory SIGN-R1
antibody. Is the deposition on these macrophages the only way B cells do see
gp120?

We have only looked at the initial interaction between B cells and gp120 bearing cells.
While our data indicates that at these early time points these macrophages provide a
major mechanism by which B cells can acquire gp120, however, we do not know if B cells
are also acquiring antigen from other cell types. Since resident and trafficking
dendritic cells should also acquire gp120 from these macrophages, they may also
participate in antigen presentation to B cells. During the time interval we have noted
the traditional subcapsular macrophages overlying the lymph node follicle capture very
little gp120, making them an unlikely source of antigen for B cells during a primary
response.

*5) It is well established that SIGN-R1 is expressed on medullary macrophages and
dendritic cells but not in interfollicular regions. A key finding of this work is
that SIGN-R1*^*+*^
*macrophages also appear to be present in interfollicular regions. It would be
important to have a more definitive examination of this. Whole lymph node imaging
providing a comparison between medullary SIGN-R1 staining and the described SIGN-R1
staining in interfollicular regions would significantly add to this work. Such an
analysis would also benefit from the addition of a CD11c marker. As recently
demonstrated by Ron Germain's group, dendritic cells in interfollicular
regions (LS DCs) are able to capture antigen; it would therefore be important to
demonstrate that SIGN-R1 interfollicular macrophages are actually capturing the gp120
and exclude the possibility that this is coming from DCs*.

We partially address this issue in Figure 5—figure supplement 2, where we show the uptake of trimeric gp120 by
SIGN-R1 positive cells overlying the interfollicular zone. We suspect that DCs are also
acquiring gp120 in the interfollicular region from the gp120 bearing SIGN-R1 positive
macrophages. We added an additional supplementary figure that shows the localization of
interfollicular CD11c^+^ cells in relation to gp120 and CD169 expressing
cells (Figure 2—figure supplement 1). In
addition, the overlay assay performed with the human lymph node section shows partial
co-localization between CD163, gp120, and DC-Sign.

*6) The authors show at the beginning of the manuscript that gp120 co-localizes
with SIGN-R1. From then onwards, they mostly rely on only the localization of gp120
and not SIGN-R1 as well, which may lead to confusion. It is important to have this
control in all the figures*.

We tried to show the important controls in each figure. In the revised manuscript, we
have added two new supplementary figures that show gp120, SIGN-R1, and a macrophage
marker. In addition, we show the human lymph node section with gp120, DC-SIGN, and CD163
partial co-localization. We also show a SIGN-R1, gp120 positive sinus lining macrophage
likely interacting with an endogenous B cell by intravital microscopy (Video 3).

*7) Is this the case for whole of HIV? It would be great if the authors were to
consider the injection of inactivated HIV. However, something that the authors will
have to deal with is to discuss their findings in the context of the wider
literature. It is important to recognize that several groups, such as the one of Uli
Von Adrian, have been making seminal contributions on virus retention in the
follicles. Similarly, the behavior of the B cells looks similar to the one described
during antigen extraction from FDCs or in vitro, which has been reported by both the
Cyster group and by the Neuberger group, the latter having actually introduced the
term*.

Should you be unable to use *HIV for these experiments you can use a more native
trimer preparation such as the ones that have been crystalized by Ian Wilson and
Peter Kwong consisting of BG505 native trimers.*

We are unable to use HIV virus; however, we were able to acquire a biotinylated trimeric
gp120 made by John Moore. When injected near the inguinal lymph node it showed a similar
binding pattern as did the monomeric gp120. Following the trimeric gp120 injection near
the inguinal lymph node, sections were prepared and the localization of gp120 determined
by staining with and labeled streptavidin. This data is shown in Figure 5—figure supplement 2.

[Editors' note: further revisions were requested prior to acceptance, as
described below.]

*In their revised manuscript the authors have added new data and made a number of
changes that help address several of the earlier concerns. However, a small number of
issues still need attention*.

*1) Data are added to show SIGN-R1+ staining of isolated LN macrophages by
fluorescence microscopy of sorted cells (*Figure 1—figure supplement 1*). However, in
the example shown it appears that 1 cell out of ∼4 is
SIGN-R1*^*+*^*. It needs to be
indicated what frequency of the sorted cells have a macrophage appearance and
SIGN-R1*^*+*^
*surface staining*.

See response to point 2.

*2) Related to 1, the amount of IFN*γ*-YFP expression by
the ‘macrophages’ is surprising given the limited evidence in the
literature for IFN*γ *expression by macrophages. Without
knowing how well excluded T cells, NK cells and ILCs are from the macrophage gate
(e.g. by frequency determination for several experiments in 1 above), it is difficult
to feel confident about the IFN*γ *data*.

We agree that the expression of IFNγ by macrophages has been controversial.
Initially we used intracellular FACS to check several different cytokines in the
“gate b” cells. Of the cytokines we checked the only one that showed a
reproducible gp120 inducible response was IFNγ. We also agree that contaminating
cells can be an issue when sorting a low frequency cell population (less than 1% in this
case). Furthermore, sorting tissue macrophage can be problematic due to their
association with other cell types. As the reviewer noted, our sorted populations were
contaminated. The issue is whether these contaminating cells gave a false positive in
our analysis. We think not, and have added some additional data to address this issue.
Using the IFNγ-YFP mouse we characterized the level of YFP expression in various
cell populations following immunization with gp120. As expected, NK and NKT cells
constitutively expressed YFP, however, gp120 immunization led to little change in their
expression levels, while a significant shift was noted after immunization in the
“gate b” cells that includes the SIGN-R1^+^ macrophages.
This can be explained by either the induction of YFP expression by the
SIGN-R1^+^ macrophages or a tight association between the
SIGN-R1^+^ macrophages and an IFNγ expressing cell type that
occurred as a consequence of the gp120 immunization. Arguing against the latter
explanation the only sorted cells that we observed to express YFP were
SIGN-R1^+^. An image of several sorted YFP positive SIGN-R1 positive
cells is shown in the Figure 1—figure supplement 3. Because of the difficulties in purifying and sorting these cells
we also used a more direct approach. We immunized the IFNγ-YFP mouse with gp120
near the inguinal lymph node and 6 hours later removed the adjacent inguinal lymph node
and as a control took a cervical lymph node. We made thick lymph node sections and
immunostained for SIGN-R1 and CD169. Confocal microscopy identified
YFP^+^ cells in the interfollicular region of the immunized LN, some
of which were SIGN-R1^+^/CD169^+/-^. In contrast, we did
not find such cells in the non-immunized cervical lymph node. Together this data
indicates that the SIGNR-1^+^ macrophages can make at least modest
levels of IFNγ following immunization with gp120.

*3) Given that the binding to
SIGN-R1*^*+*^
*cells strongly depends on glycosylation, how well matched is the glycosylation
of gp120 produced in CHO-S cells or 293F cells to the form produced in vivo by
infected T cells and macrophages? The cellular tropism of highly glycosylated
proteins may reflect more the cell types in which they are expressed than unique
properties of the antigen. In this regard, it is notable that the “two other
antigens” mentioned in the Abstract as not being captured by
SIGN-R1*^*+*^
*macrophages are also not made in CHO-S or 293F cells and neither one is a
glycoprotein. Some text should be added to the Discussion to address this issue
(perhaps commenting that the antigen studied may be most relevant to vaccine type
antigens) and to highlight the need for future studies with gp120 or HIV virions
produced from endogenously infected cell types*.

We have added some text to discuss the issues raised above.